# Few-shot Unified Question Answering: Tuning Models or Prompts?

**Srijan Bansal**♣* **Semih Yavuz**♦ **Bo Pang**♦ **Meghana Bhat**♦ **Yingbo Zhou**♦

♣ Carnegie Mellon University, ♦ Salesforce Research

srijanb@andrew.cmu.edu, {syavuz, b.pang, meghana.bhat}@salesforce.com
yingbo.zhou@salesforce.com

## Abstract

Question-answering (QA) tasks often investigate specific question types, knowledge domains, or reasoning skills, leading to specialized models catering to specific categories of QA tasks. While recent research has explored the idea of unified QA models, such models are usually explored for high-resource scenarios and require re-training to extend their capabilities. To overcome these drawbacks, the paper explores the potential of two paradigms of model tuning and prompt tuning for unified QA under a low-resource setting. The paper provides an exhaustive analysis of their applicability using 16 QA datasets spanning across various domains, revealing that prompt tuning outperforms model tuning in a few-shot setting with good initialization for out-of-distribution target tasks. The study also shows that parameter-sharing results in superior few-shot performance, and simple knowledge transfer techniques for prompt initialization can be effective in a low-resource regime. The research offers insights into the advantages and limitations of prompt tuning for unified QA in a few-shot setting, contributing to the development of effective and efficient systems in low-resource scenarios.

## 1 Introduction

Question answering (QA) is a pivotal area of research in NLP that evaluates the language understanding and reasoning capabilities of language models. To this end, the NLP community has developed numerous QA datasets that span various domains, question-answer formats, and reasoning skills (Rogers et al., 2022). Consequently, there is an increasing demand for a Unified QA system that can manage mixed batches of instances from different datasets and tasks during training and inference (Liu et al., 2022). Such a system would eliminate the need for manual tuning or per-task adjustments,

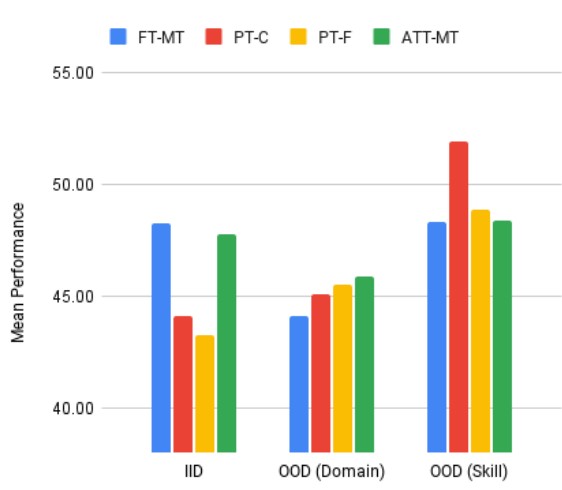

Figure 1: Comparison of Multi-Task Model-Tuning (FT-MT), Prompt-Tuning with a format based (PT-F) and a format agnostic (PT-C), and ATTEMPT (ATT-MT) (Asai et al., 2022) (a complex prompt transfer learning approach) for Unified QA on target QA datasets in a 32-shot scenario using T5-Base as the backbone model. The results show that prompt-tuning with prior outperforms multi-task full-model fine-tuning and ATTEMPT does not provide any additional advantage, especially for out-of-domain target tasks. Here IID has been used to refer to In-Distribution while OOD refers to out-of-distribution.

enabling seamless integration of new datasets. This would contribute to the development of efficient QA models with minimal computational and storage costs, enhanced generalization capabilities, and greater practicality for real-world use cases.

The success of transformer-based models in text-to-text generation has led to a growing interest in Unified QA systems. Khashabi et al. (2020) proposed Unified-QA, a single QA model pre-trained on diverse datasets that outperforms format-specialized models. While prompt-tuning methods (Lester et al., 2021; Vu et al., 2022) have emerged as a promising alternative to fine-tuning, (Zhong et al., 2022a) proposed to model the commonal-

---

*work done during internship at Salesforce Research

ities and distinguish task differences through a structurally designed prompt-based input schema. However, these approaches have limitations related to scalability, expensive pre-training requirements, and the need for tens of thousands of training examples for each task. Moreover, the performance of pre-trained QA models significantly degrades when only a few question-answering examples are available (Ram et al., 2021). While Unified QA approaches have shown success in high-data scenarios, their efficacy in more practical scenarios with limited training examples remains unexplored.

This paper aims to explore the potential of two different paradigms of tuning, model, and prompts, for unified question answering under a low resource setting. Despite the importance of this problem, there have been no previous studies investigating the effectiveness of these paradigms for this task. In response, we conduct an exhaustive analysis of the applicability of these two paradigms to a unified question-answering system. To do so, we evaluate their promise, effectiveness, and trade-offs using a set of 16 QA datasets, covering diverse domains and a wide range of skills and formats.

Our empirical study reveals several key findings, including (i) prompt tuning with good initialization can outperform model tuning under a low resource regime for out-of-distribution tasks, (ii) parameter-sharing results in superior few-shot performance, but the trends are reversed in the full-shot setting, (iii) simple knowledge transfer techniques for prompt initialization can be as effective as more complex methods in the few-shot setting, without introducing additional parameters, and (iv) prompt tuning achieves a significant performance boost from pre-training in a low resource regime while increasing model size does not significantly affect prompt tuning with initialization. In addition, we perform a systematic quantitative and qualitative study to provide insights into the advantages and limitations of prompt tuning for unified QA with an emphasis on the behaviors in the few-shot setting. Overall, our research aims to contribute to the development of effective and efficient unified question-answering systems in low-resource scenarios.

## 2 Related Work

**Parameter-efficient tuning.** Large-scale pre-trained language models fine-tuned on specific target datasets have shown remarkable performance for several downstream tasks in NLP (Devlin et al., 2019; Liu et al., 2019; Raffel et al., 2022; Brown et al., 2020; He et al., 2021b; Lan et al., 2019; Yang et al., 2019). However, standard fine-tuning approaches update all the model parameters, which can often lead to deployment challenges. Recent research (Houlsby et al., 2019; He et al., 2021c; Lester et al., 2021; Li and Liang, 2021a) has shown that similar performance can be obtained by updating or adding a few trainable parameters while keeping pre-trained language model parameters frozen. Several approaches have been proposed in this direction: Adapter-based methods (Houlsby et al., 2019; mahabadi et al., 2021; Rücklé et al., 2021) insert small trainable feed-forward networks (modules) between layers of pre-trained language models while BitFit (Ben Zaken et al., 2022) updates only the language model biases. Another computationally efficient approach is *prompt-tuning* (Lester et al., 2021) and *prefix-tuning* (Li and Liang, 2021a), which concatenate trainable continuous embeddings to the input. These trainable parameters, called soft prompts, can be used as plug-ins with a frozen LM to capture task-specific, domain-specific, or language-specific knowledge. He et al. (2021a) presents a unified view of different parameter-efficient training (PET) approaches.

**Multi-task transfer learning.** Efficient task transferability in NLP has been extensively studied (Wang et al., 2019; Liu et al., 2021a; Vu et al., 2020, 2021). With T5 (Raffel et al., 2022) demonstrating the capabilities of using existing downstream task datasets to learn a new task, proposing efficient methodologies for unifying NLP models has become a promising research paradigm in the community. Following this, (Khashabi et al., 2020) proposed UnifiedQA, a single QA model pre-trained on datasets involving diverse formats and reasoning skills. Transfer learning has been demonstrated to be effective from rich data sources (Phang et al., 2018), between similar target tasks (Vu et al., 2020), and for tasks that require similar reasoning skills (Pruksachatkun et al., 2020). However, this approach would require updating/retraining the model on a new task or a different domain, which could lead to catastrophic forgetting (Kirkpatrick et al., 2017). Moreover, Aghajanyan et al. (2021) showed approaches towards unifying NLP models suffer from negative interference to less represented tasks and between dissimilar tasks.

Most recently, Liu et al. (2022) validates that

parameter-efficient tuning methods can perform well with mixed task batches. Zhong et al. (2022b) takes the first step towards building unified QA models utilizing structural prompt tuning. Along these lines, Vu et al. (2022); Asai et al. (2022) integrates both the paradigms of parameter-efficient tuning and unifying NLP models to propose a single pre-trained model for different downstream tasks by learning target task-specific prompts from the source task prompts. Asai et al. (2022) demonstrates transfer using the attention module, while Vu et al. (2022) facilitates prompt transfer by learning the target prompt initialized from similar source prompts. These approaches require fewer than 0.1% of trainable LM parameters with little trade-off in performance.

**Few-shot question answering.** Ram et al. (2021) has identified a discrepancy between current pre-training objectives and QA, as standard models perform poorly when fine-tuned with few examples. They propose recurring span selection as a pretraining scheme tailored for question answering. Chada and Natarajan (2021), on the other hand, proposes a fine-tuning framework aligned with the pretraining framework.

However, there have been no studies focusing on the viability of prompt tuning for unified QA under low-resource settings. To address this gap, we follow prior works, (Liu et al., 2022; Asai et al., 2022; Khashabi et al., 2020), and extensively study the viability and trade-offs of prompt tuning and prompt-based transfer learning in comparison to approaches that involve full-model fine-tuning for few-shot unified QA. As a result of our comprehensive experiments, we offer essential guidelines in the form of valuable insights into the advantages and limitations of prompt tuning with respect to model tuning for unified QA in both full and few-shot scenarios.

## 3 Candidates for universal QA approach

Finetuning pre-trained language models (FT) on specific datasets yields specialized models that cater to individual tasks. However, a more efficient approach is to build a unified QA model that can perform multiple tasks without manual tuning or per-task adjustments. One of the significant advantages of such approaches is that they seamlessly support mixed-task batch inference (Liu et al., 2022), where a single model can handle diverse tasks, reducing computation, storage, and

| Approach | Paradigm | SM | KT | NT | Ex |
|---|---|---|---|---|---|
| FT | Model-tuning | | Limited | High | |
| FT-MT | Model-tuning | ✓ | High | Limited | |
| PT-R | Prompt-tuning | ✓ | Limited | Limited | ✓ |
| PT-F/PT-C | Prompt-tuning | ✓ | High | Limited | ✓ |
| ATT-MT | Prompt-tuning | ✓ | High | High | ✓ |

Table 1: Comparison of various model-tuning and prompt-tuning approaches concerning their effectiveness in fulfilling the desired properties of a unified QA model, such as Single Model (SM), Knowledge Transfer (KT), Minimal Negative Transfer (NT), and Extensibility (Ex). The table uses a checkmark symbol (✓) to indicate "Yes" and "No" for each property, and the degree of the property is indicated as "Limited" or "High" where applicable.

maintenance costs.

This study seeks to assess the suitability of two prevalent training paradigms for NLP, namely model-tuning and prompt-tuning, as potential approaches for developing a unified question-answering (QA) model. Our investigation centers around four essential criteria we look for in an effective unified QA model: (1) the ability to utilize a single model to address a range of different QA tasks, (2) effective knowledge transfer from multiple relevant tasks, (3) while minimizing the risk of negative interference, and (4) extensibility to new tasks without requiring expensive retraining. In this study, our goal is to investigate the potential of soft prompt-tuning extensively and to better understand its benefits and drawbacks in comparison with model-tuning-based approaches for building a unified QA system grounded on the aforementioned four principles. In particular, we further center the study around understanding these trade-offs in the few-shot learning scenarios, which is a realistic and more practical challenge.

**Model-tuning** This paradigm involves the fine-tuning of all the parameters of a language model to cater to a specific task or a set of tasks. Although fine-tuning (FT) on a particular dataset is an effective strategy, it is not suitable for unified QA because it requires specialized models for each dataset during inference, which is counter-intuitive to the concept of a unified QA model.

In contrast, multi-task learning via fine-tuning (FT-MT) (Raffel et al., 2022; Aribandi et al., 2021) involves the joint learning of a single model on multiple datasets by sharing all the trainable model parameters across different tasks. By training on multiple datasets, FT-MT allows for knowledge transfer

from relevant tasks during inference. However, sharing all the parameters often leads to negative transfer from unrelated tasks. Incorporating additional tasks into existing models requires retraining the model with all previous tasks and the new ones, making them computationally expensive to scale and more prone to negative interference.

**Prompt-tuning** This paradigm involves learning soft-prompt tokens added to the input while the backbone language model remains frozen. We follow the approach proposed by Lester et al. (2021) to train soft prompts for each task, where prompts are initialized from random words in the vocabulary (PT-R). This vanilla prompt-tuning approach is parameter-efficient and easy to scale. Since task-specific knowledge is captured in a different set of parameters (i.e., the prompts), this approach avoids negative interference to a great extent. With a single backbone model, we can use these prompts for different tasks. However, this approach does not leverage knowledge from other tasks not already captured in the backbone model.

Prompt initialization is a technique that addresses the issue of knowledge transfer from source tasks in vanilla prompt-tuning while retaining the benefits of a single model, minimal negative transfer, and extensibility. Previous studies (Li and Liang, 2021b; Liu et al., 2023; Vu et al., 2022) have shown that prompt-tuning methods are often sensitive to initialization, particularly in low data settings. However, the impact of different initialization methods on QA datasets has not been well studied. Inspired by (Vu et al., 2022), we initialize the target prompt by taking the average of the top-3 source task prompts most similar to the prompt trained on the target dataset. We employ two distinct approaches to this initialization process: (i) selecting source task prompts with the same answer format as that of the target dataset (PT-F), and (ii) selecting source task prompts from the complete set of source prompts (PT-C).

Apart from prompt initialization, another way to transfer knowledge from multiple tasks is through the composition of their corresponding prompts. To this end, Asai et al. (2022) proposes AT-TEMPT, a transfer learning method that learns new task-specific target prompts by computing weighted combinations of source prompts using a sub-network-based attention module trained on a single or set of tasks. We distinguish between two settings: ATT-MT, where attention modules are shared across tasks and trained in a multi-task manner, and ATT-ST, where attention module parameters are not shared. While ATT-MT provides a single model for transferring knowledge from source prompts and is easily scalable to new target tasks, sharing attention modules across tasks may result in some negative transfer, compared to more straightforward prompt-tuning methods.

## 4   Datasets

In their recent study, Rogers et al. (2022) highlight a significant increase in the number of question-answering and reading comprehension datasets, spanning various domains, formats, and reasoning abilities. This study aims to evaluate and fine-tune a range of models, leveraging a collection of datasets referred to as "source datasets" for pre-training, and a distinct set of datasets known as "target datasets" for evaluation. This paper includes datasets that cover a wide range of reasoning skills and complex linguistic phenomena, including conversational, temporal, causal, and co-reference reasoning, among others, enabling a more comprehensive evaluation of training paradigms on question-answering datasets and facilitating analysis of cross-skill transfer. This broader coverage across reasoning skills not only enables a more thorough evaluation of training paradigms on QA datasets but also facilitates analysis of cross-skill transfer. Table 3,4 presents an overview of the datasets employed in our study, detailing their size, domain, and associated primary reasoning skill.

**Source Datasets.** This study leverages source datasets for two primary purposes: pre-training models through model tuning and training source prompts via prompt-tuning approaches. The source datasets employed in our research comprise over 30,000 training instances. They aim to encompass essential reasoning skills such as reading comprehension, conversational and commonsense reasoning, as well as discrete and numerical reasoning necessary for question answering. Source datasets cover a wide range of domains, including knowledge bases, news, web documents, and Wikipedia.

**Target Datasets.** We employ target datasets to fine-tune models using the model-tuning paradigm, or to train target prompts for prompt-tuning approaches. Target datasets are typically small in size, containing fewer training instances.
**In-Distribution** : Includes datasets that share the

Table 2: Question Answering (QA) datasets used as source and target datasets in this study. For each dataset, the table provides details on associated reasoning skills, domain, and the number of training examples available. RC stands for reading comprehension

| Reasoning Skill | Dataset | Domain | Format |
|---|---|---|---|
| Reading Comprehension (RC) | SQuAD (Rajpurkar et al., 2016) | Wikipedia | EXT |
| | SearchQA (Dunn et al., 2017) | J! Archive | EXT |
| | NewsQA (Trischler et al., 2017) | news articles | EXT |
| | TriviaQA (Joshi et al., 2017) | Web, Wikipedia | EXT |
| | Natural Questions (Kwiatkowski et al., 2019) | Wikipedia | EXT |
| | NQOpen (Lee et al., 2019) | Wikipedia | ABS |
| | RACE (Lai et al., 2017) | Exams | MCQ |
| | DuoRC (Saha et al., 2018) | movie plots | |
| | PubMedQA (Jin et al., 2019) | Pubmed | MCQ |
| RC (Long) | NarrativeQA (Kočiský et al., 2018) | books, movies | ABS |
| RC (Multihop) | HotpotQA (Yang et al., 2018) | Wikipedia | EXT |
| Conversational | CoQA (Reddy et al., 2019) | News,Wiki,Books | ABS |
| | QuAC (Choi et al., 2018) | Wikipedia | EXT |
| Commonsense | ReCORD (Zhang et al., 2018) | news | ClozeQA |
| | SIQA (Sap et al., 2019) | knowledge base | MCQ |
| Discrete | DROP (Dua et al., 2019) | Wikipedia | ABS |

Table 3: Question Answering (QA) datasets used as source datasets in this study. For each dataset, the table provides details on associated reasoning skills, domain, and question format including Extractive (EXT), Abstractive (ABS) and Multi-choice (MCQ) questions.

| Reasoning Skill | Dataset | Domain | Format |
|---|---|---|---|
| In-Distribution (IID) | | | |
| Reading Comprehension (RC) | IIRC (Ferguson et al., 2020) | Wikipedia | EXT |
| | MCTest (Richardson et al., 2013) | stories | MCQ |
| | BoolQ (Clark et al., 2019) | Wikipedia | MCQ |
| | ReClor (Yu et al., 2020) | Web ,book | MCQ |
| Co-reference | Quoref (Dasigi et al., 2019) | Wikipedia | EXT |
| Commonsense | McTACO (Zhou et al., 2019) | multiple | MCQ |
| Out-of-Distribution (OOD) - Domain | | | |
| Reading | TweetQA (Xiong et al., 2019) | Twitter | ABS |
| Commonsense | COSMOSQA (Huang et al., 2019) | Spinn3r/ personal narratives | MCQ |
| | PIQA (Bisk et al., 2020) | News, Encyclopedia | MCQ |
| Conversational | CommonsenseQA (Talmor et al., 2019) | Concept Net | MCQ |
| | DREAM (Sun et al., 2019) | TOEFL | |
| | ShARC (Saeidi et al., 2018) | law rule books | ABS |
| Out-of-Distribution (OOD) - Skill | | | |
| Causal | ROPES (Lin et al., 2019) | science/ Wiki | EXT |
| | OBQA (Mihaylov et al., 2018) | science books | MCQ |
| | QuaRel (Tafjord et al., 2018) | science, etc. | MCQ |
| | COPA (Gordon et al., 2012) | personal stories | MCQ |

Table 4: Question Answering (QA) datasets used as target datasets in this study. For each dataset, the table provides details on associated reasoning skills, domain, and question format including Extractive (EXT), Abstractive (ABS) and Multi-choice (MCQ) questions.

same domain and reasoning skills as one or more of the source datasets. Examples of such datasets include MCTest and BoolQ, which cover generic domains and reasoning skills such as Wikipedia reading comprehension, respectively.

**Out-of-Distribution** : Includes datasets that lack domain knowledge and reasoning skills found in one or more of the source datasets. This subset includes datasets that involve intricate and specialized reasoning abilities such as temporal commonsense, causal reasoning, logical and inferential reasoning, as well as datasets specific to domains like Twitter, TOEFL, law books, and personal narratives. These target datasets can benefit from generic source tasks.

In some contexts, certain tasks require multiple types of reasoning. For instance, the ShARC dataset necessitates a combination of conversational and causal reasoning, while the COPA dataset entails the application of commonsense causal reasoning. Therefore, natural language processing models may face additional challenges in performing these tasks due to the integration of multiple reasoning skills. To assess the effectiveness of a unified QA system, we perform experiments on the test set of the target datasets.

## 5 Experiments

We employ the T5-base model for all our experiments, unless stated otherwise. Source prompts are trained independently for each task, while the pre-trained language model (PrLM) and attention modules for ATTEMPT are trained jointly on all the source tasks. For target datasets, we randomly select a small number of instances for few-shot training and evaluation. The hyperparameters for training are presented in section 5.1. Table 7 details the initialization used for different target tasks in both PT-F and PT-C. We select the best checkpoint based on the validation set performance, with FT-MT and ATT-MT using a single validation set comprising of all the target tasks, and PT-R, PT-F, and PT-C using a validation set for each target task individually. We evaluate the best checkpoint on the test set of each target dataset using F1 as the metric for extractive and abstractive QA datasets, and accuracy for MCQ and Yes/No QA datasets. In cases where a test set is unavailable, we use the development set to report our model's performance and create a small subset from the training set for hyperparameter tuning and checkpoint se-

| Backbone : T5-Base | | | | | | | |
|---|---|---|---|---|---|---|---|
| k-shot | FT | FT-MT | PT-R | PT-F | PT-C | ATT-ST | ATT-MT |
| 16 | 45.12 | **45.62** | 38.92 | 44.61 | 45.55 | 45.70 | 45.23 |
| 32 | 46.36 | 47.00 | 42.76 | 45.75 | 46.88 | 47.68 | **47.40** |
| 64 | 48.08 | 49.12 | 44.34 | **49.33** | 48.85 | 48.56 | 49.08 |
| 128 | 50.14 | **52.27** | 44.41 | 50.31 | 50.83 | 48.76 | 50.31 |
| 256 | 52.96 | **55.39** | 45.77 | 53.15 | 52.31 | 50.21 | 52.68 |
| 512 | 56.60 | **59.54** | 47.03 | 54.91 | 55.34 | 51.20 | 54.70 |

Table 5: Comparison of Model-Tuning and Prompt-Tuning Paradigms in Few-Shot setting: Model-tuning approaches include FT and FT-MT, while PT-R represents vanilla prompt tuning and PT-F and PT-C correspond to prompt tuning with initialization. ATT-ST and ATT-MT are single-task and multi-task variants of ATTEMPT, a prompt transfer learning approach. **Bold** values indicate the best model with a T5-base backbone for the k-shot scenario, while underline represents the second-best.

lection. We report the aggregate results of three seeds. Table 5 summarizes the experimental results comparing the model-tuning and prompt-tuning paradigms for a unified QA system. In the rest of this section, we share our key findings and insights that can hopefully help guide which paradigm to prefer under which scenarios.

## 5.1 Hyper-parameters

After extensive tuning, we selected a learning rate of 1e-5 for the backbone model, along with a maximum source length of 512, a gradient accumulation step of 2, and a batch size of 16. During training, we saved and evaluated checkpoints every 500 steps, and trained the model for 100K steps with patience. For all experiments, the prompts consisted of k = 100 tokens with a hidden dimension of d = 768.

**Best Candidate for Unified QA** FT-MT, PT-R, PT-F, PT-C, ATT-MT are potential candidates for the unified question answering task. In low-resource scenarios, all candidates perform similarly, but PT-F and PT-C stands out due to its low number of trainable parameters and ease of scaling to new datasets. As the number of training instances increases, FT-MT outperforms other approaches, while prompt-tuning approaches remain competitive. Our findings suggest that a simple approach like PT-F is on par with more sophisticated prompt-transfer learning approaches like ATT-MT.

Table 6 presents a comparison between model-tuning and prompt-tuning techniques in the 32-shot and 64-shot settings for various categories. The term *IID* refers to in-distribution target tasks such as BoolQ, IIRcm MCTest, ReClor, MCTACO, and Quoref. On the other hand, *OOD (Domain)* encompasses TweetQA, COSMOSQA, CSQA, DREAM, and ShARC, while *OOD (skill)* includes ROPES,

|  | IID | OOD (Domain) | OOD (Skill) |
|---|---|---|---|
| 32 Examples | | | |
| FT-MT | **48.27** | 44.12 | 48.34 |
| PT-C | 44.14 | 45.08 | **51.95** |
| PT-F | 43.27 | 45.56 | 48.89 |
| ATT-MT | 47.82 | **45.90** | 48.41 |
| 64 Examples | | | |
| FT-MT | 50.62 | 46.15 | 50.28 |
| PT-C | 49.76 | 45.04 | **51.54** |
| PT-F | 51.52 | 46.30 | 49.75 |
| ATT-MT | 50.05 | 46.98 | 50.02 |

Table 6: Comparison of model-tuning and prompt-tuning paradigms in few-shot setting for different categories of target tasks.

QuaRel, PIQA, OBQA, and COPA. It is observed that prompt-based techniques significantly outperform multi-task model tuning when dealing with out-of-distribution target tasks. However, for in-distribution tasks, ATT-MT performs comparably to FT-MT. These results indicate the superiority of simple prompt initialization methods in terms of adaptability in few-shot scenarios compared to model tuning. Complex prompt transfer methods like ATTEMPT have demonstrated their usefulness in IID tasks exclusively. Importantly, PT-C outperforms Pt-F for new skills, suggesting the transferability of tasks beyond format boundaries.

**Parameter-sharing results in superior few-shot performance; however, the trends are reversed in the full-shot setting.** Multi-task fine-tuning (FT-MT) that employs the parameter-sharing technique yields superior few-shot learning performance than traditional finetuning (FT). The extent of improvement increases with the number of training examples and starts decreasing at a threshold

of approximately 512 examples on an aggregate level. However, this can vary across different target datasets. For instance, datasets such as TweetQA, PIQA, and ReClor exhibit this behavior beyond 512 examples, while OBQA, MCTest, and CommonsenseQA realize this at around 128. Increasing training examples from 16 to 512 leads to a boost from $\Delta = 0.50$ to $\Delta = 2.94$ due to transfer learning from other tasks. However, raising the number of examples to 1024 results in a drop in improvement to $\Delta = 1.77$. In the full-shot setting with unbalanced training samples, employing parameter sharing among all tasks can lead to negative interference, resulting in a reversal of the trend where FT outperforms FT-MT. Similarly, sharing attention modules across all target tasks in multi-task prompt transfer learning approaches (ATT-MT) leads to a comparable trend.

**Format-based prompt initialization achieves comparable performance to more complex prompt-transfer approaches.** The Prompt-tuning paradigm has emerged as a highly effective approach for fine-tuning pre-trained language models for specific tasks. However, it has been shown that the success of this paradigm can be highly sensitive to initialization. To address this issue, we drew inspiration from the work of (Vu et al., 2022) and explored the use of two different initialization techniques for the target prompt (PT-F and PT-C). Our results demonstrated that both initialization techniques outperformed random initialization by 6% with 32 examples, and this gap increased to approximately 20% with 1024 examples. Notably, we found that the simpler format-based heuristic initialization was just as effective as the more complex cosine-based search over the entire prompt pool. Furthermore, our results revealed that both prompt initialization approaches were competitive with the sophisticated attention-module-based prompt-transfer learning approach ATT-MT.

Our analysis further revealed that the performance of PT-F and PT-C varied based on the skill or domain of the dataset (see Table 10). Evaluation on datasets from specific domains (Figure 9) reveals that in low-regime scenarios, PT-F outperformed PT-C in the Web+Social and domain-specific book domains, while PT-C was more effective for Knowledge graphs and Wikipedia domains. However, in high-range scenarios, all models performed similarly. Furthermore, our analysis from

a skill perspective, as depicted in Figure 10, indicated that PT-F performed better in Dialog reasoning in the low range and in commonsense reasoning in the high range. On the other hand, PT-C was better suited for causal reasoning in the low range. More detailed information on our findings can be found in the appendix in Table 10.

| Target Dataset | Format-based (Pt-F) | Complete Set (PT-C) |
|---|---|---|
| ropes | searchqa, newsqa, **quac** | drop, siqa, **quac** |
| dream | record, race, **siqa** | searchqa, quac, **siqa** |
| sharc | duorc, **coqa**, **nar_qa** | quac, **coqa**, **nar_qa** |
| boolq | **pubmed_qa** | race, newsqa, **pubmed_qa** |
| piqa | record, **siqa**, race | quac, **siqa**, nar_qa |
| quoref | quac, newsqa, nq | duorc, nar_qa, drop |
| cosmos_qa | record, **siqa**, **race** | nar_qa, **siqa**, **race** |
| tweet_qa | **nq_open**, **duorc**, **nar_qa** | **nq_open**, **duorc**, **nar_qa** |
| CQA | record, race, **siqa** | newsqa, nar_qa, **siqa** |
| obqa | record, **race**, **siqa** | newsqa, **race**, **siqa** |
| reclor | record, race, **siqa** | duorc, **siqa**, nq_open |
| quarel | record, race, siqa | duorc, quac, nar_qa |
| mctest | record, **race**, **siqa** | nq, **race**, **siqa** |
| mc_taco | **pubmed_qa** | **pubmed_qa**, siqa, nar_qa |
| copa | pubmed_qa | searchqa, race, siqa |
| iirc | hotpotqa, **newsqa**, **nq** | **newsqa**, drop, **nq** |

Table 7: Source Prompts most similar to target prompts for format-based and complete-set initialization corresponding to PT-F and PT-C respectively. **Bold** indicates source tasks common in both partitions. Although some source prompts are shared across target tasks, Quoref and COPA have none in common. The SIQA and RACE source prompts are typically used for initialization, but we found that lifting the constraint of choosing prompts from the same format allowed for successful cross-format initialization at the reasoning skill or domain level. For example, the DREAM dataset (MCQ) was initialized with QuAC (ExtQA), which is reasonable since both involve conversational data. The IIRC dataset was also initialized with the most relevant source task, HotpotQA. Yes/No questions strongly prefer PubmedQA as a format.

## 6 Discussion

### 6.1 Qualitative Analysis

**Do different models agree on their answers?** Fig 7 shows the average agreement of different models on all the tasks across different few-shot scenarios. We find that PT-C and PT-F have the highest agreement scores. We partly attribute this to the high overlap of initialization prompts of format and logically similar tasks (PT-C, PT-C). However, as the number of shots increases the overall agreement decreases across different modes. Furthermore, we investigate if different modes can be complementary to each other by evaluating the union of their predictions across different shots. We find that fine-

tuning (FT) and model tuning models (FT-MT) are complementary to each other at low resource settings whereas the gains from PT-R to other modes are minimum. For the complete results, refer to Appendix (Figure 3). This might indicate that prompt tuning may not be practical without good initialization for extremely low-resource QA scenarios. For further discussions around few shot analysis, refer to Appendix 6.2.

**A closer look at the task-level performance across different few-shot settings reveals counter-intuitive behaviors.** We find that under low resource settings (< 256 shot) good initialization helps significantly for target tasks that are similar to source tasks (e.g: OBQA, BoolQ, IIRC), and the performance gain decreases as we increase the number of shots. As seen from Figure 8, for similar tasks PT-C and model tuning FT-MT performed significantly better than PT-R. However, in cases where there is little domain overlap (ShaRC), initializations do not contribute substantially to the overall performance of the model. Interestingly, in some cases, we find counter-intuitive results where performance remains flat (ShaRC) from Figure 8) or zig-zag (Ropes) pattern is observed across different shots. We point the reader to Appendix (Figures 4, 5, 8) for performance across different modes against different shots.

## 6.2 Qualitative Study

Table 8 presents a few qualitative examples across different shots and modes. We find prompt tuning with good initialization to leverage world knowledge better (e.g: Arctic Circle with cold weather) even in low resource settings while prompt tuning struggles in predicting local context-based reasoning tasks (e.g: taking photos of home does not associate with new home).

**Do the same model across different few-shot settings agree on its answers?** Figure **??** presents the overall agreement of different models for a single task under different shot settings. We observe patterns of high level agreement between adjacent shots that gradually decrease with an increase in the number of shots in fine-tuning and prompt tuning with initialization mode. However, prompt tuning with random initialization has an agreement percentage of 50% across different shots and has no clear distinction of high agreement between the adjacent shots as found in other settings.

## 7 Conclusion

In this work, we explore the viability of prompt-tuning as a solution to unified QA and conduct a thorough analysis of its promise, effectiveness, and trade-offs compared with the model-tuning paradigm on a set of 16 QA datasets, focusing particularly on several few-shot scenarios. As a result, we obtain several key findings and insights that hopefully will inform which paradigm to prefer under which scenarios. Prompt tuning is quite competitive with model-tuning in the lower extreme of the few-shot scenarios, given a good initialization. While parameter-sharing leads to superior performance in the few-shot setting, the trends flip in the full-shot setting, A simple knowledge transfer approach (i.e., an average of relevant prompts) is as effective as complex methods without introducing additional parameters. Pre-training the backbone model on the source tasks significantly benefits prompt tuning. While initializing from a strong prior is very helpful for prompt tuning, its benefit is not as substantial when using a larger backbone model, especially when the number of training examples exceeds a certain threshold.

## Limitations

Our work has several limitations: (1) since few-shot experiments are prone to have considerable variance due to the randomly sampled few training examples, we repeat all the experiments using three randomness seeds for the T5-base backbone. However, since the number of experiments per seed is more than 1500, we were able to run the same experiments with a T5-large backbone using only one seed and excluding specific few-shot settings due to computational limitations, especially given the latter model has 3.5 times more parameters. Although our comparisons of the two models are still presented in an entirely fair fashion using the same single seed, it would have been more strongly conclusive to test our findings with a T5-base backbone on the larger model to the same extent. That is also the reason why the current version of our study does not include comparisons with even larger models such as T5-3b or T5-11b. (2) We explore a limited number of prompt-tuning methods both in terms of how the soft prompts are injected in the model architecture following (Lester et al., 2021) and how the knowledge from source tasks are used to inform target tasks following (Vu et al., 2022; Asai et al., 2022). For example, Liu

(2022) proposes a parameter-efficient fine-tuning alternative to soft prompt-tuning in recent work, while (Zhong et al., 2022a) shows the benefits of prompt-based pretraining. Although the key takeaways in the current version of our study are supported by sufficient empirical evidence, incorporating the aforementioned recent developments may prove even further promise and evidence for the prompt-based approaches towards few-shot unified QA. (3) Our study is currently limited to English-QA datasets, hindering our findings to be generally valid for cross-lingual and/or cross-model question-answering systems. Therefore, we need to consider how our findings would generalize to other languages and modalities.

## Ethical Statement

We observe a preference for multiple-choice question (MCQ) answer formats across various question-answering (QA) datasets with varying levels of reasoning ability. Additionally, the majority of the source datasets were sourced from Wikipedia, which may contain gender or political bias that could be further perpetuated by models. The T5 model, which was used for pre-training, may also have biases due to its pre-training data. However, the study did not conduct stress tests to identify potential biases, and users should be cautious when implementing the provided models. The current models' results may not align with the facts in input documents, potentially leading to the spread of false information online. This is a common issue in all current QA models, and further research is needed in this area. The study's experiments were primarily conducted using A100 GPUs and consumed a significant amount of GPU time when repeated across random seeds. Nevertheless, our findings can benefit subsequent studies and applications by providing valuable insights, thus avoiding the need for extensive repetition of these comparisons.

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

## A  Appendix

### A.1  Effect of Pre-training

**Pre-training improves performance in few-shot scenarios, particularly in the lower range, with significant benefits observed in prompt-tuning.** Following Unified-QA (Khashabi et al., 2020), we observe that pre-training the T5-base model on diverse source datasets with varying formats and skill requirements (as shown in Table 2) can boost the performance of the pre-trained language model (PrLM) in both fine-tuning and prompt-tuning scenarios. Our analysis reveals that pre-training can yield substantial performance gains through knowledge transfer from source tasks, especially when few training examples are available (refer to Figure 1). We further observe that prompt-tuning with a pre-trained LM introduces inductive bias in prompts, resulting in a much greater performance boost than FT-MT, with the difference becoming more pronounced as the number of instances increases (potentially due to overfitting). Specifically, PT-R yields a change in improvement from 36% to 24% as the number of training instances increases from 16 to 1024, while improvement in FT-MT drastically reduces from 27% to 7%. We note that ATT-MT follows a similar pattern to that of Model Tuning (MT). Moreover, our findings indicate that datasets such as COSMOSQA, OBQA, DREAM, MCTest, IIRC, and BoolQ exhibit substantial performance gains through pre-training, likely due to their similarity to some of the source datasets. On the other hand, datasets such as McTACO, QuaRel, ShARC, and PIQA, which are less closely related to the source datasets, do not exhibit significant improvements with pre-training.

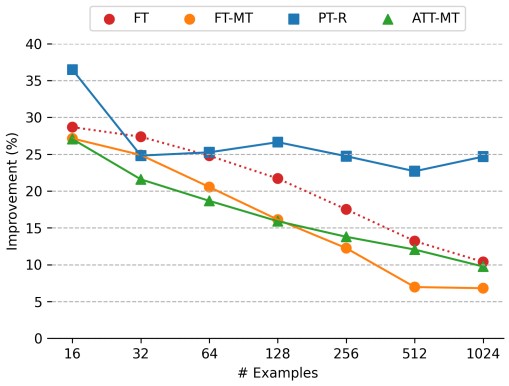

Figure 2: Improvement (%) of pre-trained backbone model (PrLM) over T5-base observed across different training approaches in few-shot setting.

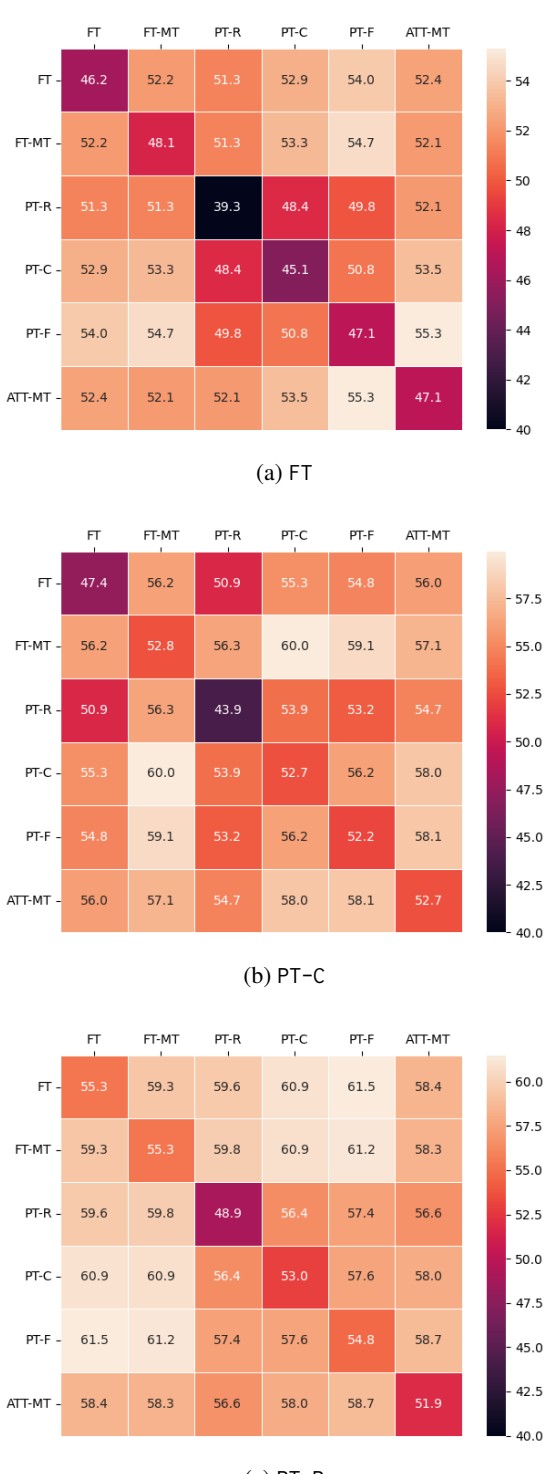

(a) FT

(b) PT-C

(c) PT-R

Figure 3: Heatmaps showing union matrix of different shots for each mode

| Example | # Few Shot | Label | FT-MT | PT-C | PT-R |
|---|---|---|---|---|---|
| **Context:** M: How long have you been teaching? W: To be frank, I'm tired of teaching the same textbook though I do enjoy being a teacher. I'm considering trying something new. **Question:** What's the woman probably going to do? **Options:** (A) To teach a different book. (B) To change her job. (C) To learn a different textbook. | 128 | **(B)** | **(B)** | (A) | (C) |
| **Context:** Q: Are you a Native American/American Indian? A: Yes; I just finished high school. My parents wanted me to go to college, but I never applied. **General Instructions:** ... who has been accepted or enrolled in an accredited degree program, in the field of health care, and you or your family member. **Question:** Do I qualify for this benefit program? | 256 | **No** | **No** | Yes | Yes |
| **Context:** cold temperatures cause animals to shiver **Question:** Where would animals shiver the most? **Options:** (A) Arctic Circle (B) Sumatra (C) Java (D) tropical rainforest | 512 | **Arctic Circle** | Sumatra | **Arctic Circle** | Java |
| **Context:** The house painters finished...While I would not say they were not the greatest guys... they did do a nice job and the house looks so much better. Here are some photos ... **Question:** What may have caused you to take photos of your house? **Options:** (A) It got a new coat of color. (B) It was my new house. (C) I wanted to show off its old coat of color. (D) None of the above choices. | 128 | **(A)** | **(A)** | (B) | (B) |

Table 8: Table presenting qualitative examples showing model predictions across different shots for different tasks. Few shot column shows the shot until which the predictions in the table hold.

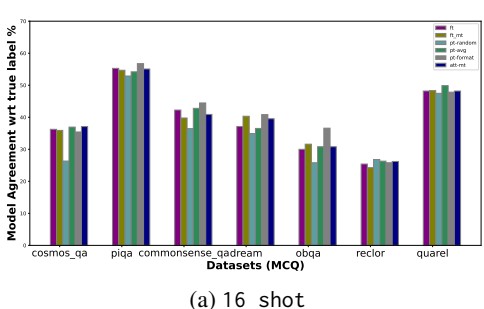

(a) 16 shot

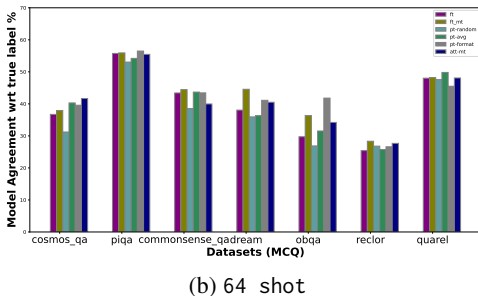

(b) 64 shot

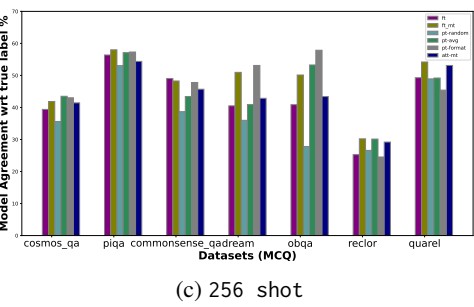

(c) 256 shot

Figure 4: Graphs showing task level agreement across different shots for different modes

## B  Variation with model size

| k-shot | size | FT | MT | PT (R) | PT (B) | ATT |
|---|---|---|---|---|---|---|
| 16 Examples | Base | 43.68 | 45.31 | 38.72 | 45.30 | 45.33 |
| | Large | 52.19 | 53.74 | 46.34 | 50.21 | 50.96 |
| | Δ | 8.50 | 8.44 | 7.63 | 4.92 | 5.63 |
| 32 Examples | Base | 45.94 | 47.11 | 42.04 | 47.47 | 46.81 |
| | Large | 55.96 | 56.86 | 48.09 | 50.35 | 51.16 |
| | Δ | 10.02 | 9.75 | 6.04 | 2.88 | 4.34 |
| 128 Examples | Base | 49.91 | 51.22 | 44.01 | 50.65 | 50.56 |
| | Large | 60.25 | 61.94 | 50.01 | 52.07 | 51.76 |
| | Δ | 10.34 | 10.72 | 6.00 | 1.42 | 1.20 |
| 1024 Examples | Base | 59.55 | 60.68 | 46.14 | 57.00 | 59.51 |
| | Large | 69.71 | 69.87 | 52.37 | 58.06 | 57.15 |
| | Δ | 10.16 | 9.19 | 6.23 | 1.06 | -2.35 |

Table 9: Comparison between model-tuning and prompt-tuning paradigms for different model sizes. Mean performance over 16 target datasets is reported with T5 as a pre-trained language model.

Recent studies have shown that the performance gap between prompt-tuning and fine-tuning reduces as the model size increases (Liu et al., 2021b). In this work, we conduct experiments comparing the performance of base vs large variants of T5 for a range of different fine-tuning methods as shown in Table 9. Unless otherwise specified, we use the T5-base model for our experimentation. We observe a consistent improvement in performance with large language models. Specifically, model-tuning approaches achieve a consistent improvement of approximately 10 points across 32 to 1024 training instances, while prompt-tuning without initialization achieves an improvement of roughly 6 points. However, prompt-tuning with initialization and ATTEMPT do not show significant improvement in performance with large models, and this improvement diminishes as the number of training instances increases. The limited performance gain from large models leads us to conclude that multi-task model-tuning outperforms prompt-tuning and

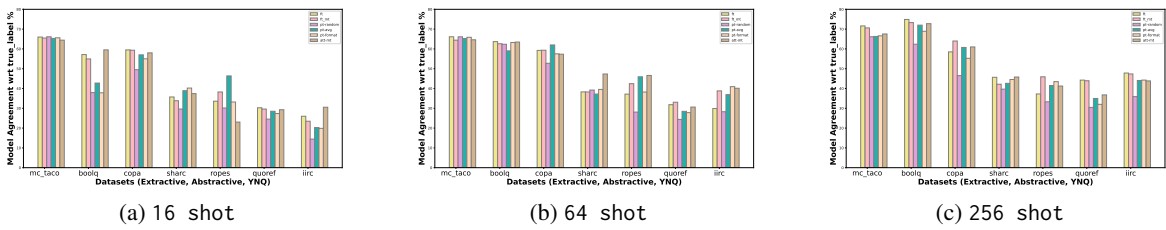

(a) 16 shot  (b) 64 shot  (c) 256 shot

Figure 5: Graphs showing task level agreement across different shots for different modes

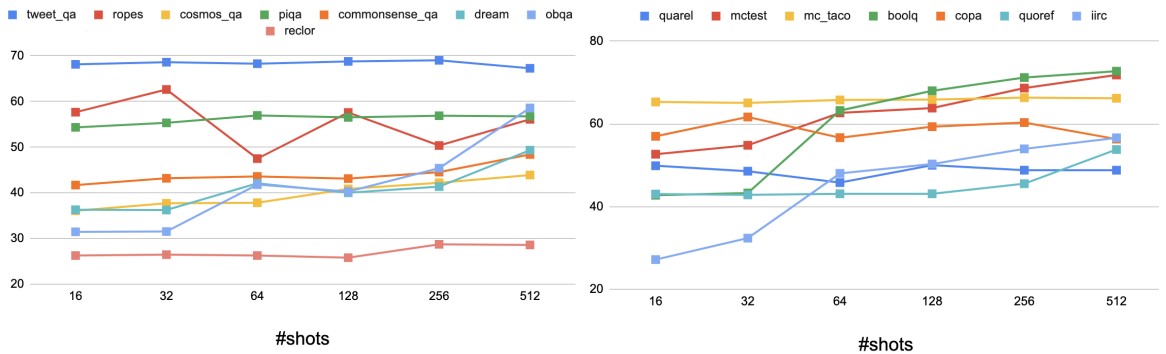

Figure 6: Graphs showing performance of PT-C on target datasets for different shots.

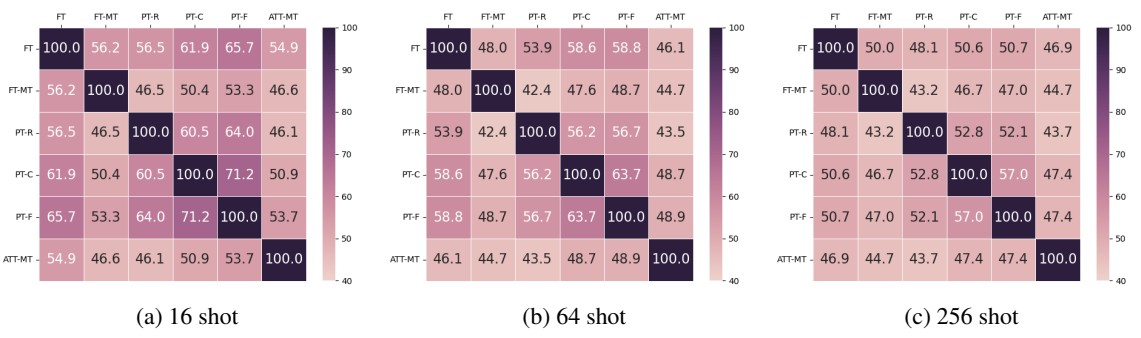

(a) 16 shot  (b) 64 shot  (c) 256 shot

Figure 7: Heatmaps showing agreement matrix of different modes under different few shot settings

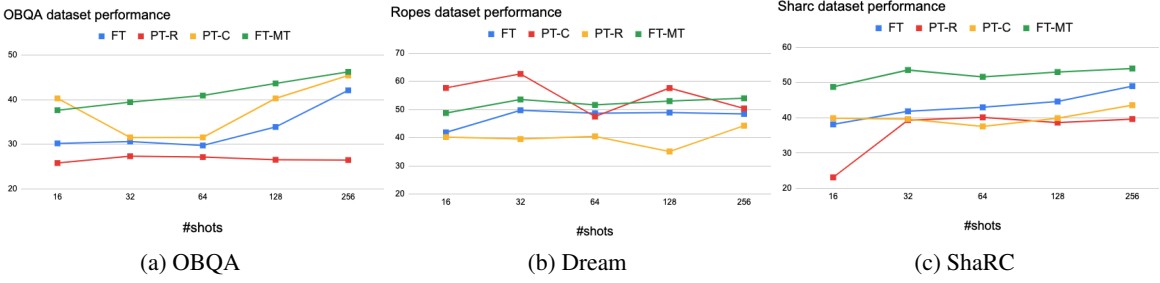

(a) OBQA  (b) Dream  (c) ShaRC

Figure 8: The effect of prior and training data size on certain target tasks

ATTEMPT-MT in a few-shot setting. Nevertheless, prompt-tuning with initialization remains comparable to ATTEMPT-MT, and both methods significantly outperform prompt-tuning without initialization. Overall, these findings suggest that the effectiveness of different candidates for unified QA may depend on the size of the model and the number of training instances available.

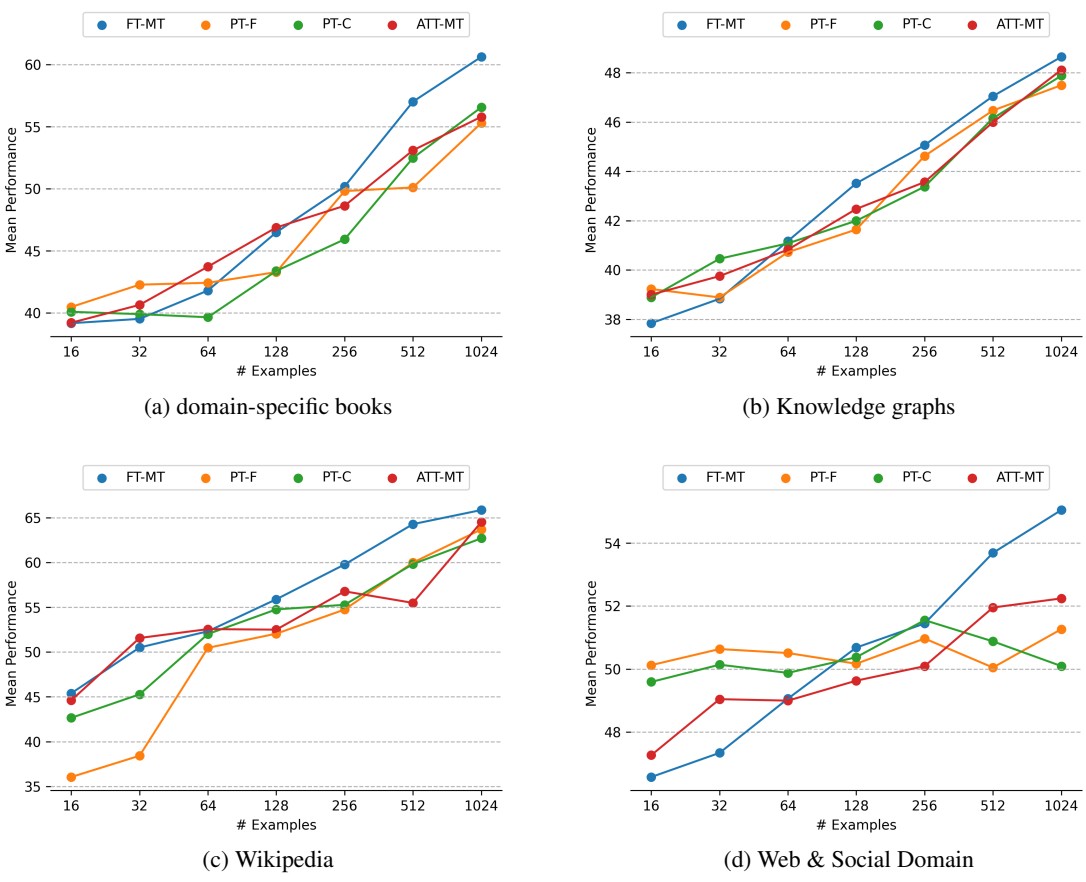

Figure 9: Comparison of FT-MT, PT-F, PT-C and ATT-MT in several few-shot scenarios using T5-Base as the backbone model for different domains

| Skill-based | Domain-based |
|---|---|
| **Machine Reading Comprehension** | **Web & Social Media** |
| iirc, tweet_qa, mctest, boolq, reclor | tweet_qa, piqa, reclor |
| **Commonsense Reasoning** | **Wikipedia** |
| cosmos_qa, piqa, commonsense_qa, mc_taco | iirc, ropes, quoref, boolq |
| **Dialog Reasoning** | **Knowledge Graph** |
| sharc, dream, quoref | commonsense_qa, cosmos_qa |
| **Causal Reasoning** | **domain specific book** |
| ropes, obqa, quarel, copa | sharc, obqa, quarel |

Table 10: Categorization of Target Datasets Based on Domain and Reasoning Skill

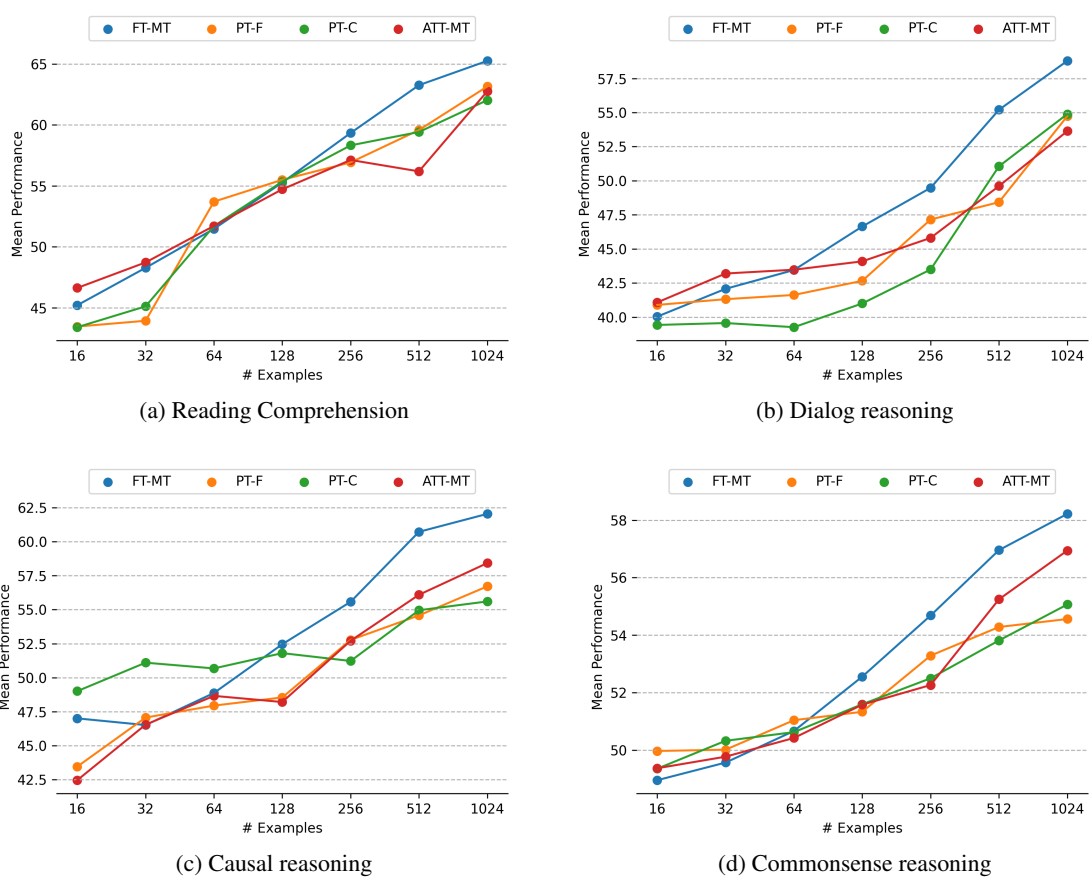

Figure 10: Comparison of FT-MT, PT-F, PT-C and ATT-MT in several few-shot scenarios using T5-Base as the backbone model for different reasoning skills

| Model | tweet_qa | ropes | cosmos_qa | piqa | CQA | dream | obqa | reclor | sharc | quarel | mctest | mc_taco | boolq | copa | quoref | iirc | agg |
|---|---|---|---|---|---|---|---|---|---|---|---|---|---|---|---|---|---|
| *16 Examples* | | | | | | | | | | | | | | | | | |
| FT | 65.26 | 41.87 | 35.00 | 55.51 | 41.63 | 37.52 | 30.20 | 25.27 | 38.13 | 48.08 | 52.33 | 66.03 | 50.70 | 58.00 | 43.64 | 32.80 | 45.12 |
| FT-MT | 60.77 | 48.76 | 35.90 | 54.68 | 39.78 | 40.33 | 31.60 | 24.27 | 37.62 | 48.32 | 50.33 | 65.48 | 54.93 | 59.33 | 42.17 | 35.72 | 45.62 |
| PT-R | 39.97 | 40.22 | 25.93 | 52.39 | 35.68 | 35.92 | 25.80 | 25.80 | 23.10 | 47.48 | 53.50 | 66.13 | 37.90 | 55.00 | 42.34 | 23.22 | 38.92 |
| PT-F | 68.15 | 37.04 | 35.10 | 56.09 | 43.35 | 40.67 | 34.53 | 26.13 | 39.68 | 47.24 | 58.17 | 65.36 | 37.83 | 55.00 | 42.34 | 27.00 | 44.61 |
| PT-C | 68.15 | 57.67 | 36.07 | 54.35 | 41.71 | 36.31 | 31.47 | 26.27 | 38.94 | 49.88 | 52.67 | 65.31 | 42.75 | 57.00 | 43.03 | 27.18 | 45.55 |
| att-st | 63.38 | 31.94 | 34.82 | 54.91 | 40.57 | 36.90 | 32.93 | 27.20 | 39.17 | 50.12 | 54.00 | 66.20 | 51.12 | 61.67 | 45.80 | 40.49 | 45.70 |
| ATT-MT | 60.52 | 32.82 | 37.10 | 55.08 | 40.90 | 39.54 | 30.80 | 26.20 | 38.65 | 48.20 | 45.83 | 64.42 | 59.51 | 58.00 | 45.05 | 41.06 | 45.23 |
| *32 Examples* | | | | | | | | | | | | | | | | | |
| FT | 65.55 | 49.74 | 36.08 | 55.42 | 41.85 | 37.43 | 30.60 | 24.33 | 41.84 | 48.08 | 52.17 | 66.14 | 60.52 | 60.67 | 44.77 | 26.56 | 46.36 |
| FT-MT | 61.01 | 53.55 | 36.05 | 55.68 | 41.61 | 42.53 | 32.40 | 25.33 | 39.42 | 46.76 | 50.83 | 64.95 | 58.20 | 53.33 | 44.28 | 46.00 | 47.00 |
| PT-R | 35.72 | 39.50 | 32.23 | 53.01 | 35.76 | 36.00 | 27.33 | 25.27 | 39.34 | 46.52 | 51.33 | 66.06 | 61.80 | 53.67 | 41.25 | 39.35 | 42.76 |
| PT-F | 68.61 | 46.41 | 34.80 | 56.22 | 42.97 | 41.44 | 40.20 | 27.07 | 40.00 | 46.64 | 59.17 | 66.09 | 37.83 | 55.00 | 42.51 | 26.97 | 45.75 |
| PT-C | 68.61 | 62.64 | 37.72 | 55.33 | 43.19 | 36.24 | 31.53 | 26.47 | 39.63 | 48.56 | 54.83 | 65.08 | 43.28 | 61.67 | 42.83 | 32.37 | 46.88 |
| att-st | 63.80 | 50.97 | 34.52 | 55.11 | 41.88 | 36.88 | 33.20 | 27.20 | 44.18 | 49.64 | 53.50 | 66.31 | 53.96 | 64.67 | 46.46 | 40.69 | 47.68 |
| ATT-MT | 64.11 | 55.53 | 36.94 | 55.89 | 42.56 | 41.55 | 30.87 | 27.13 | 44.33 | 46.76 | 45.33 | 63.72 | 59.82 | 53.00 | 43.72 | 47.20 | 47.40 |
| *64 Examples* | | | | | | | | | | | | | | | | | |
| FT | 66.17 | 48.65 | 36.73 | 55.53 | 43.65 | 38.42 | 29.73 | 24.93 | 42.97 | 48.08 | 52.67 | 66.13 | 65.31 | 60.67 | 46.85 | 42.83 | 48.08 |
| FT-MT | 62.94 | 51.62 | 37.91 | 55.91 | 44.44 | 44.56 | 36.33 | 28.33 | 40.90 | 48.20 | 53.33 | 64.42 | 62.61 | 59.33 | 44.90 | 50.13 | 49.12 |
| PT-R | 57.15 | 40.45 | 31.65 | 53.12 | 37.51 | 36.57 | 27.13 | 25.93 | 40.12 | 47.48 | 53.00 | 66.12 | 62.39 | 53.67 | 39.11 | 38.03 | 44.34 |
| PT-F | 68.29 | 47.50 | 37.84 | 56.96 | 43.60 | 42.09 | 41.80 | 26.27 | 39.69 | 45.80 | 62.67 | 65.78 | 63.24 | 56.67 | 43.10 | 48.04 | 49.33 |
| PT-C | 68.29 | 58.97 | 38.43 | 55.01 | 43.73 | 37.21 | 31.53 | 26.33 | 37.56 | 49.88 | 57.83 | 65.49 | 61.22 | 63.67 | 43.02 | 44.82 | 48.85 |
| att-st | 63.26 | 53.20 | 37.45 | 55.89 | 42.56 | 37.14 | 32.80 | 26.27 | 43.11 | 49.04 | 53.50 | 65.61 | 58.27 | 63.67 | 46.96 | 48.25 | 48.56 |
| ATT-MT | 63.86 | 55.02 | 41.68 | 55.46 | 39.97 | 40.51 | 34.20 | 27.67 | 48.91 | 48.08 | 52.83 | 64.60 | 63.47 | 57.33 | 41.01 | 50.73 | 49.08 |
| *128 Examples* | | | | | | | | | | | | | | | | | |
| FT | 66.42 | 48.92 | 40.76 | 55.22 | 46.44 | 37.71 | 33.87 | 25.33 | 44.62 | 47.96 | 53.00 | 67.64 | 72.10 | 57.33 | 50.14 | 54.79 | 50.14 |
| FT-MT | 64.91 | 52.98 | 41.03 | 57.00 | 46.00 | 47.89 | 43.33 | 30.13 | 43.62 | 52.52 | 59.17 | 66.21 | 67.75 | 61.00 | 48.43 | 54.38 | 52.27 |
| PT-R | 58.26 | 35.09 | 32.73 | 53.23 | 37.10 | 36.27 | 26.53 | 26.07 | 38.65 | 49.04 | 52.83 | 64.94 | 62.15 | 54.33 | 41.53 | 41.87 | 44.41 |
| PT-F | 68.79 | 47.89 | 39.03 | 56.58 | 44.25 | 43.99 | 42.33 | 25.13 | 41.26 | 46.28 | 66.00 | 65.49 | 64.90 | 57.67 | 42.74 | 52.66 | 50.31 |
| PT-C | 68.79 | 57.61 | 40.86 | 56.53 | 43.13 | 40.02 | 40.27 | 25.80 | 39.89 | 50.00 | 63.83 | 65.88 | 68.01 | 59.33 | 43.10 | 50.30 | 50.83 |
| att-st | 63.48 | 52.39 | 38.32 | 54.99 | 42.70 | 36.67 | 32.93 | 26.60 | 44.76 | 50.24 | 53.50 | 66.21 | 61.45 | 57.67 | 47.33 | 50.85 | 48.76 |
| ATT-MT | 65.34 | 46.84 | 42.14 | 54.28 | 42.81 | 42.27 | 41.33 | 29.27 | 47.99 | 51.32 | 57.83 | 67.11 | 69.23 | 53.33 | 42.01 | 51.91 | 50.31 |
| *256 Examples* | | | | | | | | | | | | | | | | | |
| FT | 67.64 | 48.42 | 42.00 | 55.82 | 49.39 | 40.51 | 42.07 | 25.07 | 48.99 | 49.28 | 61.83 | 71.33 | 74.78 | 58.67 | 52.92 | 58.62 | 52.96 |
| FT-MT | 66.13 | 53.98 | 41.86 | 58.00 | 48.27 | 50.96 | 50.13 | 30.20 | 46.21 | 54.20 | 66.50 | 70.61 | 73.29 | 64.00 | 51.27 | 60.55 | 55.39 |
| PT-R | 63.26 | 44.25 | 36.36 | 53.39 | 38.08 | 36.24 | 26.47 | 27.00 | 39.65 | 49.16 | 55.50 | 66.14 | 62.31 | 46.67 | 43.08 | 54.77 | 45.77 |
| PT-F | 69.03 | 51.80 | 42.27 | 57.40 | 46.98 | 50.96 | 57.93 | 26.47 | 45.46 | 46.04 | 67.00 | 66.51 | 67.34 | 55.33 | 45.05 | 54.77 | 53.15 |
| PT-C | 69.03 | 50.37 | 42.21 | 56.89 | 44.55 | 41.36 | 45.40 | 28.73 | 43.58 | 48.80 | 68.67 | 66.36 | 71.21 | 60.33 | 45.54 | 53.96 | 52.31 |
| att-st | 62.79 | 54.77 | 39.79 | 55.91 | 46.25 | 37.34 | 36.67 | 26.40 | 46.70 | 49.40 | 51.67 | 66.09 | 65.73 | 63.00 | 47.57 | 53.34 | 50.21 |
| ATT-MT | 66.68 | 53.33 | 41.43 | 54.39 | 45.70 | 42.86 | 43.40 | 29.20 | 49.36 | 53.12 | 61.17 | 67.55 | 72.75 | 61.00 | 45.21 | 55.80 | 52.68 |
| *512 Examples* | | | | | | | | | | | | | | | | | |
| FT | 68.11 | 54.84 | 45.45 | 56.91 | 51.76 | 45.08 | 53.40 | 26.73 | 53.11 | 48.44 | 69.17 | 74.11 | 76.47 | 61.67 | 57.89 | 62.39 | 56.60 |
| FT-MT | 67.98 | 58.25 | 42.75 | 58.74 | 51.35 | 56.23 | 60.00 | 34.33 | 52.02 | 58.99 | 72.50 | 74.99 | 75.46 | 65.67 | 57.36 | 66.02 | 59.54 |
| PT-R | 65.44 | 45.15 | 36.49 | 53.66 | 38.74 | 34.20 | 33.73 | 27.47 | 40.16 | 48.44 | 54.33 | 66.07 | 62.51 | 51.33 | 45.06 | 49.77 | 47.03 |
| PT-F | 67.28 | 54.33 | 43.96 | 57.13 | 48.98 | 52.92 | 61.33 | 25.73 | 40.65 | 48.32 | 71.00 | 67.07 | 75.64 | 54.33 | 51.72 | 58.23 | 54.91 |
| PT-C | 67.28 | 56.10 | 43.91 | 56.75 | 48.40 | 49.35 | 58.60 | 28.60 | 50.03 | 48.80 | 71.83 | 66.25 | 72.74 | 56.33 | 53.80 | 56.63 | 55.34 |
| att-st | 61.56 | 55.95 | 39.35 | 55.51 | 46.74 | 40.26 | 43.47 | 27.73 | 47.60 | 50.00 | 56.17 | 66.09 | 68.73 | 57.67 | 48.32 | 54.04 | 51.20 |
| ATT-MT | 68.23 | 58.65 | 44.32 | 56.29 | 47.67 | 46.72 | 53.80 | 31.33 | 51.87 | 53.60 | 68.33 | 72.71 | 50.52 | 58.33 | 50.24 | 62.52 | 54.70 |
| *1024 Examples* | | | | | | | | | | | | | | | | | |
| FT | 68.90 | 52.90 | 46.21 | 56.46 | 54.57 | 52.17 | 60.60 | 27.93 | 56.12 | 52.52 | 75.50 | 78.76 | 76.99 | 65.00 | 63.15 | 67.55 | 59.71 |
| FT-MT | 70.59 | 58.57 | 42.91 | 58.74 | 54.38 | 59.22 | 64.80 | 35.80 | 55.87 | 61.15 | 76.33 | 76.84 | 76.27 | 63.67 | 61.28 | 67.26 | 61.48 |
| PT-R | 64.58 | 39.94 | 35.88 | 52.97 | 40.87 | 36.03 | 33.60 | 26.53 | 42.41 | 46.64 | 53.67 | 66.54 | 62.15 | 46.67 | 46.98 | 52.16 | 46.73 |
| PT-F | 67.39 | 55.21 | 42.66 | 56.40 | 52.33 | 52.19 | 62.80 | 30.00 | 53.53 | 49.52 | 77.33 | 66.86 | 76.45 | 59.33 | 58.50 | 64.57 | 57.82 |
| PT-C | 67.39 | 51.05 | 44.46 | 56.49 | 51.30 | 51.86 | 63.93 | 26.40 | 54.28 | 51.44 | 75.17 | 68.02 | 74.51 | 56.00 | 58.53 | 66.67 | 57.34 |
| att-st | 69.69 | 57.27 | 40.22 | 55.77 | 50.01 | 47.60 | 55.27 | 26.87 | 53.74 | 49.64 | 62.50 | 66.59 | 75.02 | 60.67 | 50.83 | 59.28 | 55.06 |
| ATT-MT | 69.46 | 59.06 | 44.88 | 56.13 | 51.35 | 50.15 | 59.00 | 31.13 | 54.03 | 54.32 | 71.00 | 75.41 | 75.52 | 61.33 | 56.74 | 66.69 | 58.51 |
| *Full* | | | | | | | | | | | | | | | | | |
| FT | 77.40 | 59.16 | 69.82 | 67.63 | 62.74 | 66.32 | 74.20 | 47.20 | 67.36 | 67.99 | 78.00 | 99.41 | 82.97 | 67.00 | 71.18 | 70.57 | 70.56 |
| FT-MT | 76.34 | 58.53 | 67.30 | 67.03 | 61.18 | 67.40 | 74.20 | 36.20 | 63.35 | 59.35 | 73.50 | 92.21 | 80.58 | 66.00 | 70.19 | 69.88 | 67.70 |
| PT-R | 75.63 | 55.32 | 55.58 | 60.94 | 59.38 | 60.49 | 68.40 | 40.60 | 58.72 | 62.23 | 74.00 | 95.95 | 80.61 | 55.00 | 68.48 | 69.20 | 65.03 |
| PT-F | 73.31 | 51.31 | 49.78 | 58.11 | 56.59 | 62.65 | 71.20 | 35.00 | 57.24 | 56.12 | 78.50 | 78.87 | 79.54 | 59.00 | 64.10 | 68.64 | 62.50 |
| PT-C | 75.23 | 52.13 | 58.69 | 60.93 | 58.07 | 64.12 | 71.40 | 41.60 | 58.53 | 62.95 | 76.50 | 96.43 | 79.76 | 62.00 | 67.58 | 69.55 | 65.97 |
| att-st | 55.31 | 60.29 | 59.45 | 79.33 | 62.62 | 68.68 | 61.94 | 75.71 | 58.56 | 70.22 | 69.00 | 97.90 | 66.91 | 77.50 | 64.00 | 42.60 | 66.88 |
| ATT-MT | 74.37 | 57.02 | 58.63 | 61.53 | 58.89 | 61.67 | 67.80 | 39.60 | 60.11 | 57.19 | 77.00 | 94.19 | 80.86 | 65.00 | 67.71 | 68.89 | 65.65 |

Table 11: Complete set of results for comparison between model-tuning and prompt-tuning approaches on 16 target QA datasets with T5-base as pre-trained language model.

| | Backbone : T5-Base | | | | | | | Backbone : PrLM | | | |
|---|---|---|---|---|---|---|---|---|---|---|---|
| k-shot | FT | MT | PT-R | PT-F | PT-B | ATT-ST | ATT-MT | FT | MT | PT-R | ATT-MT |
| 16 | 4.07 | 2.74 | 6.05 | 1.45 | 2.18 | 3.49 | 3.29 | 1.84 | 2.11 | 4.18 | 2.24 |
| 32 | 1.28 | 2.71 | 6.11 | 2.90 | 3.11 | 3.50 | 2.85 | 1.68 | 1.60 | 2.77 | 1.85 |
| 64 | 4.60 | 2.53 | 2.02 | 3.29 | 3.03 | 2.42 | 3.21 | 1.13 | 1.87 | 2.35 | 2.42 |
| 128 | 2.25 | 2.22 | 2.12 | 4.13 | 4.01 | 1.73 | 2.76 | 1.34 | 1.39 | 2.80 | 2.81 |
| 256 | 3.36 | 2.40 | 1.97 | 3.25 | 4.23 | 3.12 | 3.63 | 1.41 | 1.90 | 2.84 | 2.93 |
| 512 | 2.62 | 1.35 | 4.22 | 1.35 | 2.04 | 3.77 | 2.45 | 0.83 | 1.22 | 2.23 | 3.17 |
| 1024 | 1.73 | 1.71 | 4.31 | 2.62 | 2.21 | 3.95 | 2.93 | 1.35 | 2.09 | 2.78 | 2.01 |

Table 12: Table displays the aggregate standard deviation of target tasks with different seeds. Increasing training instances reduces standard deviation, improving model robustness and reducing sensitivity to minor variations. PrLM reduces standard deviation across all approaches, leading to stable performance and better generalization while addressing overfitting. Prompt tuning has a higher deviation due to initialization sensitivity. Parameter-sharing and prompt initialization techniques reduce deviation, leveraging knowledge from other tasks for stable performance, especially in low-resource scenarios, and mitigating overfitting.

| Model | tweet_qa | ropes | cosmos_qa | piqa | CQA | dream | obqa | reclor | sharc | quarel | mctest | mc_taco | boolq | copa | quoref | iirc | agg |
|---|---|---|---|---|---|---|---|---|---|---|---|---|---|---|---|---|---|
| *16 Examples* | | | | | | | | | | | | | | | | | |
| FT | 73.17 | 50.92 | 47.52 | 55.79 | 52.33 | 68.02 | 67.47 | 35.80 | 40.14 | 47.12 | 86.50 | 67.03 | 73.59 | 68.00 | 49.88 | 45.75 | 58.06 |
| FT-MT | 71.44 | 52.13 | 47.18 | 57.44 | 51.02 | 68.61 | 67.40 | 37.27 | 40.68 | 48.68 | 84.83 | 67.40 | 76.15 | 65.33 | 48.61 | 43.90 | 58.00 |
| PT-R | 61.75 | 52.32 | 42.64 | 55.97 | 40.92 | 59.00 | 59.07 | 36.27 | 40.36 | 48.20 | 78.17 | 66.13 | 68.71 | 56.33 | 48.20 | 35.83 | 53.12 |
| ATT-MT | 69.37 | 49.36 | 47.07 | 56.09 | 52.47 | 65.78 | 66.60 | 36.40 | 41.61 | 50.12 | 85.83 | 67.14 | 71.67 | 67.67 | 47.09 | 45.18 | 57.47 |
| *32 Examples* | | | | | | | | | | | | | | | | | |
| FT | 73.32 | 51.69 | 48.61 | 55.77 | 52.69 | 68.02 | 67.87 | 36.27 | 42.45 | 48.92 | 86.67 | 66.23 | 74.22 | 70.33 | 51.08 | 50.59 | 59.05 |
| FT-MT | 71.46 | 52.67 | 46.58 | 57.02 | 51.00 | 68.64 | 68.47 | 36.80 | 42.94 | 50.00 | 84.67 | 67.41 | 79.18 | 63.67 | 50.53 | 48.21 | 58.70 |
| PT-R | 61.64 | 51.26 | 43.27 | 56.13 | 41.96 | 63.68 | 63.07 | 35.60 | 39.79 | 47.96 | 80.83 | 66.14 | 65.48 | 52.67 | 49.42 | 35.02 | 53.37 |
| ATT-MT | 70.09 | 50.39 | 48.94 | 55.79 | 51.73 | 66.23 | 67.67 | 35.80 | 38.81 | 48.68 | 86.17 | 66.29 | 72.30 | 65.00 | 51.26 | 46.89 | 57.63 |
| *64 Examples* | | | | | | | | | | | | | | | | | |
| FT | 74.17 | 51.56 | 49.39 | 55.59 | 53.26 | 68.69 | 68.80 | 35.73 | 43.21 | 47.48 | 86.50 | 66.51 | 80.88 | 66.33 | 53.92 | 58.00 | 60.00 |
| FT-MT | 72.06 | 52.55 | 48.15 | 56.78 | 52.17 | 68.82 | 67.73 | 37.87 | 40.10 | 49.16 | 84.83 | 66.70 | 80.10 | 64.00 | 54.09 | 52.26 | 59.21 |
| PT-R | 66.83 | 53.82 | 46.91 | 55.53 | 42.37 | 63.15 | 64.20 | 36.13 | 40.76 | 48.20 | 80.50 | 66.22 | 74.07 | 58.33 | 49.93 | 41.75 | 55.54 |
| ATT-MT | 71.61 | 51.48 | 48.14 | 55.98 | 52.33 | 66.06 | 68.33 | 36.07 | 40.15 | 50.36 | 85.67 | 63.57 | 77.32 | 66.33 | 52.39 | 46.09 | 58.24 |
| *128 Examples* | | | | | | | | | | | | | | | | | |
| FT | 74.00 | 51.63 | 50.73 | 56.51 | 54.79 | 68.30 | 69.00 | 35.80 | 43.89 | 48.44 | 86.33 | 67.71 | 81.70 | 66.67 | 56.16 | 64.60 | 61.02 |
| FT-MT | 72.69 | 51.04 | 49.96 | 57.18 | 54.00 | 69.12 | 68.13 | 37.60 | 44.25 | 52.52 | 85.33 | 68.38 | 81.53 | 63.67 | 56.61 | 59.04 | 60.69 |
| PT-R | 71.70 | 54.12 | 45.62 | 56.35 | 42.34 | 66.01 | 63.87 | 36.47 | 39.19 | 48.08 | 83.83 | 65.81 | 72.94 | 58.33 | 50.79 | 44.39 | 56.24 |
| ATT-MT | 71.76 | 52.32 | 48.39 | 56.18 | 51.05 | 66.86 | 68.80 | 34.73 | 40.33 | 50.36 | 85.33 | 62.20 | 78.50 | 59.00 | 54.49 | 52.64 | 58.31 |
| *256 Examples* | | | | | | | | | | | | | | | | | |
| FT | 74.00 | 52.58 | 52.23 | 56.20 | 55.88 | 69.26 | 69.67 | 37.53 | 46.86 | 49.76 | 86.83 | 69.75 | 82.02 | 65.00 | 60.49 | 67.60 | 62.23 |
| FT-MT | 72.28 | 53.47 | 50.97 | 58.14 | 54.05 | 69.87 | 67.53 | 40.20 | 48.83 | 54.44 | 82.83 | 70.85 | 81.73 | 65.33 | 59.20 | 64.95 | 62.17 |
| PT-R | 71.71 | 52.73 | 48.15 | 55.75 | 47.94 | 67.19 | 65.40 | 34.27 | 39.75 | 48.32 | 84.17 | 65.81 | 75.92 | 55.00 | 50.96 | 50.33 | 57.09 |
| ATT-MT | 71.41 | 52.20 | 49.65 | 56.33 | 52.06 | 66.21 | 69.73 | 37.73 | 44.07 | 52.52 | 86.33 | 67.96 | 78.50 | 62.67 | 55.05 | 56.64 | 59.94 |
| *512 Examples* | | | | | | | | | | | | | | | | | |
| FT | 74.34 | 55.50 | 53.45 | 58.54 | 57.41 | 68.84 | 70.67 | 36.40 | 55.78 | 51.08 | 86.67 | 75.79 | 82.14 | 65.33 | 63.96 | 69.18 | 64.07 |
| FT-MT | 72.79 | 55.85 | 52.74 | 58.89 | 55.26 | 70.18 | 69.40 | 39.13 | 53.94 | 54.92 | 85.50 | 71.93 | 81.61 | 66.67 | 62.65 | 67.49 | 63.68 |
| PT-R | 71.98 | 54.72 | 48.08 | 55.84 | 48.38 | 66.67 | 69.93 | 34.60 | 37.24 | 50.48 | 84.33 | 66.15 | 74.74 | 60.33 | 51.66 | 47.99 | 57.70 |
| ATT-MT | 73.27 | 53.55 | 51.66 | 55.91 | 51.13 | 62.53 | 67.93 | 37.00 | 49.49 | 55.04 | 85.17 | 70.23 | 81.36 | 60.33 | 60.66 | 65.41 | 61.29 |
| *1024 Examples* | | | | | | | | | | | | | | | | | |
| FT | 73.83 | 57.80 | 57.20 | 58.29 | 58.75 | 70.85 | 70.67 | 42.27 | 59.91 | 53.48 | 87.33 | 80.24 | 82.20 | 66.00 | 66.28 | 69.26 | 65.90 |
| FT-MT | 73.87 | 59.29 | 54.84 | 59.65 | 56.21 | 70.39 | 71.13 | 43.33 | 58.16 | 62.59 | 83.33 | 79.16 | 81.27 | 64.00 | 65.04 | 68.25 | 65.66 |
| PT-R | 71.79 | 52.84 | 48.59 | 56.13 | 51.11 | 66.63 | 68.53 | 34.27 | 38.71 | 49.40 | 85.33 | 65.70 | 77.19 | 61.33 | 55.15 | 49.38 | 58.26 |
| ATT-MT | 74.78 | 59.91 | 54.42 | 55.93 | 51.82 | 67.21 | 70.67 | 39.87 | 56.24 | 56.95 | 86.50 | 75.09 | 81.00 | 65.67 | 62.51 | 68.78 | 64.21 |
| *Full* | | | | | | | | | | | | | | | | | |
| FT | 79.05 | 65.52 | 71.06 | 67.85 | 62.16 | 73.24 | 72.20 | 54.40 | 66.99 | 73.74 | 85.00 | 99.40 | 82.75 | 69.00 | 72.78 | 72.19 | 72.96 |
| FT-MT | 77.16 | 61.79 | 69.82 | 67.25 | 61.75 | 73.33 | 72.20 | 42.60 | 65.94 | 69.42 | 83.00 | 93.37 | 82.91 | 69.00 | 71.61 | 70.62 | 70.74 |
| PT-R | 77.04 | 62.60 | 66.57 | 61.70 | 60.85 | 70.69 | 70.60 | 42.80 | 59.38 | 70.86 | 85.50 | 95.59 | 82.35 | 61.00 | 71.28 | 70.84 | 69.35 |
| ATT-MT | 77.34 | 63.56 | 67.14 | 63.22 | 61.26 | 70.20 | 68.80 | 45.40 | 60.82 | 69.42 | 84.50 | 89.65 | 82.05 | 65.00 | 72.13 | 70.67 | 69.45 |

Table 13: Complete set of results for comparison between model-tuning and prompt-tuning approaches on 16 target QA datasets with PrLM as pre-trained language model trained on source tasks.

| Model | tweet_qa | ropes | cosmos_qa | piqa | CQA | dream | obqa | reclor | sharc | quarel | mctest | mc_taco | boolq | copa | quoref | iirc | agg |
|---|---|---|---|---|---|---|---|---|---|---|---|---|---|---|---|---|---|
| *16 Examples* | | | | | | | | | | | | | | | | | |
| ft | 75.12 | 44.38 | 39.63 | 56.91 | 53.40 | 41.27 | 40.80 | 27.40 | 28.54 | 49.28 | 73.50 | 64.50 | 74.65 | 67.00 | 48.85 | 49.74 | 52.19 |
| ft-mt | 71.87 | 46.81 | 38.83 | 59.79 | 55.28 | 52.75 | 42.80 | 31.00 | 31.37 | 55.04 | 73.50 | 63.66 | 71.41 | 72.00 | 46.97 | 46.84 | 53.74 |
| pt-random | 74.79 | 54.52 | 37.09 | 55.60 | 48.32 | 41.13 | 31.80 | 22.60 | 8.46 | 48.56 | 56.50 | 66.10 | 59.54 | 58.00 | 51.04 | 27.46 | 46.34 |
| pt-best | 72.91 | 45.23 | 38.39 | 54.95 | 51.52 | 39.75 | 31.80 | 22.40 | 38.87 | 46.76 | 59.00 | 66.20 | 74.16 | 65.00 | 53.56 | 42.92 | 50.21 |
| att-mt | 69.38 | 56.38 | 36.72 | 55.82 | 52.91 | 39.51 | 29.60 | 26.80 | 41.19 | 46.76 | 65.00 | 66.14 | 65.78 | 65.00 | 54.52 | 43.83 | 50.96 |
| *32 Examples* | | | | | | | | | | | | | | | | | |
| ft | 75.56 | 57.00 | 41.24 | 54.62 | 51.76 | 48.77 | 50.40 | 26.00 | 38.49 | 50.00 | 71.00 | 67.17 | 73.18 | 76.00 | 57.88 | 56.34 | 55.96 |
| ft-mt | 72.84 | 52.79 | 41.81 | 58.65 | 55.61 | 58.53 | 52.00 | 31.20 | 39.16 | 53.96 | 71.50 | 65.99 | 70.64 | 77.00 | 54.47 | 53.63 | 56.86 |
| pt-random | 75.62 | 51.59 | 38.19 | 54.73 | 52.25 | 40.15 | 32.00 | 22.40 | 14.93 | 47.48 | 58.00 | 66.38 | 69.79 | 65.00 | 52.55 | 28.31 | 48.09 |
| pt-best | 72.82 | 46.39 | 38.22 | 55.50 | 51.43 | 39.95 | 32.00 | 21.00 | 39.43 | 47.12 | 57.50 | 66.13 | 73.39 | 65.00 | 54.19 | 45.60 | 50.35 |
| att-mt | 68.92 | 58.59 | 37.22 | 55.77 | 52.91 | 39.56 | 31.80 | 26.00 | 41.89 | 48.56 | 60.50 | 66.15 | 65.87 | 65.00 | 53.47 | 46.31 | 51.16 |
| *128 Examples* | | | | | | | | | | | | | | | | | |
| ft | 75.37 | 42.31 | 47.84 | 59.52 | 55.53 | 58.63 | 59.80 | 30.20 | 49.75 | 58.27 | 76.00 | 71.41 | 80.18 | 73.00 | 62.59 | 63.65 | 60.25 |
| ft-mt | 73.23 | 50.07 | 49.45 | 62.13 | 56.92 | 64.71 | 61.60 | 34.80 | 48.92 | 58.63 | 80.50 | 74.09 | 79.54 | 73.00 | 61.20 | 62.27 | 61.94 |
| pt-random | 75.23 | 60.79 | 36.35 | 56.37 | 48.73 | 41.13 | 32.80 | 21.40 | 40.40 | 49.64 | 59.50 | 66.17 | 68.87 | 62.00 | 52.66 | 28.20 | 50.01 |
| pt-best | 73.68 | 57.55 | 37.96 | 55.11 | 52.09 | 40.44 | 29.80 | 23.80 | 40.84 | 48.92 | 59.50 | 65.97 | 76.24 | 69.00 | 55.44 | 46.74 | 52.07 |
| att-mt | 73.13 | 61.37 | 38.12 | 54.08 | 54.22 | 39.71 | 26.80 | 26.60 | 42.57 | 50.00 | 59.50 | 66.13 | 69.08 | 62.00 | 56.02 | 48.80 | 51.76 |
| *1024 Examples* | | | | | | | | | | | | | | | | | |
| ft | 78.71 | 54.74 | 56.38 | 63.44 | 66.91 | 71.67 | 73.40 | 41.80 | 59.23 | 72.66 | 85.50 | 85.89 | 83.67 | 79.00 | 70.42 | 71.99 | 69.71 |
| ft-mt | 77.17 | 54.05 | 56.98 | 64.31 | 64.05 | 74.95 | 74.60 | 42.40 | 61.20 | 72.66 | 88.00 | 85.56 | 81.90 | 80.00 | 69.36 | 70.68 | 69.87 |
| pt-random | 73.38 | 56.78 | 38.39 | 54.35 | 54.46 | 40.59 | 32.40 | 24.40 | 47.05 | 52.88 | 60.50 | 66.08 | 73.39 | 63.00 | 55.85 | 44.37 | 52.37 |
| pt-best | 73.19 | 56.41 | 35.78 | 53.92 | 59.62 | 37.40 | 57.00 | 28.00 | 54.07 | 47.84 | 81.50 | 66.45 | 81.35 | 65.00 | 67.27 | 64.24 | 58.06 |
| att-mt | 75.57 | 50.53 | 39.30 | 55.01 | 58.89 | 49.85 | 49.20 | 26.60 | 50.63 | 52.52 | 78.00 | 66.13 | 79.82 | 67.00 | 59.89 | 55.52 | 57.15 |
| *Full* | | | | | | | | | | | | | | | | | |
| ft | 81.34 | 68.10 | 77.79 | 72.80 | 72.24 | 78.73 | 81.40 | 50.60 | 71.37 | 77.34 | 88.00 | 99.48 | 86.02 | 80.00 | 76.63 | 73.33 | 77.20 |
| ft-mt | 80.31 | 64.66 | 74.67 | 71.38 | 73.46 | 78.97 | 80.60 | 49.20 | 68.00 | 72.66 | 88.50 | 93.74 | 83.98 | 78.00 | 77.12 | 73.96 | 75.58 |
| pt-random | 79.20 | 57.26 | 66.83 | 68.17 | 70.52 | 77.25 | 76.60 | 32.40 | 62.52 | 75.90 | 85.00 | 97.26 | 84.89 | 70.00 | 74.30 | 71.11 | 71.83 |
| pt-best | 80.08 | 58.00 | 67.57 | 67.36 | 68.55 | 77.40 | 78.20 | 42.80 | 63.33 | 72.30 | 86.50 | 98.28 | 83.79 | 73.00 | 75.34 | 73.43 | 72.87 |
| att-mt | 78.88 | 54.88 | 57.35 | 63.87 | 68.14 | 67.45 | 74.60 | 33.40 | 59.79 | 62.59 | 84.50 | 85.12 | 84.59 | 69.00 | 74.62 | 71.33 | 68.13 |

Table 14: Complete set of results for comparison between model-tuning and prompt-tuning approaches on 16 target QA datasets with T5-Large as pre-trained language model.

| Example | Task | label | ft_mt | uqa_mt | init_random | init_format | init_squad | att_mt |
|---|---|---|---|---|---|---|---|---|
| how do you use cream? context: (A) spray it all over your skin. (B) get a small amount of it, and spread it all over your skin wherever you'd like to apply it until you can't see it anymore. | piqa | (B) | (B) | (B) | (B) | (B) | *(A)* | (B) |
| How to keep cool outside.? context: (A) Add some peppermint oil to a spray bottle of water, spray self often avoiding eyes. (B) Sit under a shade tree. | piqa | (B) | (B) | (B) | (B) | (B) | (B) | *(A)* |
| How to grow a plant.? context: (A) Bury seed in sand and add 1 cup of water daily. (B) Bury seed in soil and add 1 cup of water daily. | piqa | (B) | (B) | (B) | (B) | (B) | (B) | (B) |
| Which sample will most likely rust first? Context: ... | ropes | Sample B | Sample A | Sample B | Sample B | Sample B | Sample B | Sample B |
| Will St. Louis have more or less acid rain than Seattle? Context:... ... St.Louis recently installed a coal-fired power plant... When acid rain falls, | ropes | more | less | less | less | more | more | less |
| Which city will more likely have acid rain? Context:... St. Louis recently installed a coal-fired power plant... When acid rain falls, | ropes | St.Louis | St.Louis | St.Louis | St.Louis | St.Louis | St.Louis | St.Louis |
| Tommy glided across the marble floor with ease, but slipped and fell on the wet floor because — has more resistance. ? | quarel | wetfloor | wetfloor | wetfloor | wetfloor | marble floor | wetfloor | wetfloor |
| Mars has a greater mass than the moon. Which object will attract fewer objects to it? | quarel | Moon | Mars | Moon | Mars | Mars | Mars | Mars |