# OpenReview forum: "Few-shot Unified Question Answering: Tuning Models or Prompts?"
_EMNLP/2023/Conference — EMNLP 2023 Findings_

### Official Review · Reviewer_Y5aV · 2023-07-26

**Typos Grammar Style And Presentation Improvements:** N/A
**Soundness:** 3

**Excitement:**

3: Ambivalent: It has merits (e.g., it reports state-of-the-art results, the idea is nice), but there are key weaknesses (e.g., it describes incremental work), and it can significantly benefit from another round of revision. However, I won't object to accepting it if my co-reviewers champion it.

**Missing References:**

1. He K, Huang Y, Mao R, et al. Virtual prompt pre-training for prototype-based few-shot relation extraction[J]. Expert Systems with Applications, 2023, 213: 118927.

The paper talk about prompt initialization and under few-shot setting.

**Paper Topic And Main Contributions:**

The paper argues that current QA models are typically designed for high-resource scenarios and require re-training to be useful in low-resource settings. To address this issue, the authors explore the potential of two paradigms: model tuning and prompt tuning, for unified QA in a low-resource setting.

The authors' results demonstrate that prompt tuning outperforms model tuning in a few-shot setting with a good initialization for out-of-distribution target tasks. Additionally, they show that parameter-sharing leads to superior few-shot performance, and that simple knowledge transfer techniques for prompt initialization can be effective in a low-resource regime.

**Questions For The Authors:**

What is the meaning of PT(R) or PT(B) in Table 6 ? I cannot find any explanations. It would be helpful add the explanation in the caption of Table 6.

**Reasons To Accept:**

1. The paper conducts a sufficient number of experiments and provides a systematic and comprehensive analysis of the results. The authors' findings are based on a rigorous evaluation of their proposed approaches and highlight the potential of prompt tuning and parameter-sharing for unified QA in low-resource settings.

2. In addition to the quality of their experiments, the paper is well-organized and easy to follow, which enhances its readability and accessibility to a broad range of readers.
While the paper lacks a comparison under larger models, the authors' goals are still meaningful and important, warranting continued research and development in this area.


**Reasons To Reject:**

1. The paper did’t compared with any Large Language Model (LLM), such as chatGPT or Claude, which lessening the impact of the paper.  I understand the paper may be finished in advance, but there are three main concerns that the comparison is matters:
    a. The investigation goal 1 “the ability to utilize a single model to address a range of different QA tasks” is the exact problem that LLM can solve.
    b. The paper aim to explore QA under  low-resource setting — The investigation goal 4 “extensibility to new tasks without requiring expensive retraining”, which are also LLM is good at.

    Although you may cannot tuning  LLM such as chatGPT, but for me, applying In-context-learning with 16/32 demonstration example is still comparable with your explored model tuning or prompt tuning. Besides, tuning a LLaMA 6B/17B are also affordable and convincing.


2. Considering the paper is empirical study, only using T5-Base/large model may weak the generalizability of you findings.  For instance, the conclusion that "parameter-sharing results in superior few-shot performance, but the trends are reversed in the full-shot setting", and the observation that "format-based prompt initialization achieves comparable performance to more complex prompt-transfer approaches", I will wonder that if the conclusion  still hold when using larger models like T5-XXl, or models with 6B size or larger. It would be interesting to investigate the robustness of these conclusions when using larger models.

**Reproducibility:**

4: Could mostly reproduce the results, but there may be some variation because of sample variance or minor variations in their interpretation of the protocol or method.

**Reviewer Confidence:**

3: Pretty sure, but there's a chance I missed something. Although I have a good feel for this area in general, I did not carefully check the paper's details, e.g., the math, experimental design, or novelty.

---

> ### Author Rebuttal · Authors · 2023-08-29
>
> Thank you for taking the time to review our work and providing your valuable feedback.
>
> >The investigation goal 1 “the ability to utilize a single model to address a range of different QA tasks” is the exact problem that LLM can solve.
>
> While LLMs have demonstrated a promising ability to adapt to various tasks using in-context examples, previous studies (https://arxiv.org/pdf/2302.06476.pdf, https://arxiv.org/pdf/2302.03494.pdf) have highlighted their limitations in handling complex spatial, temporal, and causal common sense reasoning tasks, which are covered in the scope of this study. It is evident that finetuned models significantly outperform zero-shot LLM models on tasks involving symbolic reasoning, logical reasoning, and question-answering. Since all of these complex tasks can be posed as question-answering tasks, the identified shortcomings underscore the necessity for a model capable of effectively addressing multiple complex QA datasets while maintaining competitive performance within a computational budget for real-world scenarios.
>
> >Although you may cannot tuning LLM such as chatGPT, but for me, applying In-context-learning with 16/32 demonstration example is still comparable with your explored model tuning or prompt tuning.
>
> Prior research, such as https://arxiv.org/pdf/2205.05638.pdf, indicates the superiority and cost-effectiveness of few-shot Parameter-Efficient Fine-Tuning over In-Context Learning, providing further motivation for this work. Additionally, insights derived from parameter-efficient learning for medium-sized language models should offer valuable guidance for approaches like LoRA-based finetuning (e.g., Llama), which is another popular and parameter-efficient method.
>
> > I will wonder that if the conclusion still hold when using larger models like T5-XXl, or models with 6B size or larger.
>
> Our observations from Table 6 demonstrate that prompt initialization achieves comparable performance to even more complex prompt-transfer approaches, even for the T5-Large model.
>
> >It would be interesting to investigate the robustness of these conclusions when using larger models.
>
> While this study only shows results on an encoder-decoder model like T5, it can be easily extended to other model architectures and sizes, such as 6B or larger. This work aims to evaluate the viability of PEFT-based techniques for a set of complex QA datasets. At the time of this study, there were no open-source LLMs available for training, thus we will leave these explorations for future work.
>
> >What is the meaning of PT(R) or PT(B) in Table 6 ? I cannot find any explanations. It would be helpful add the explanation in the caption of Table 6.
>
> Lines 242-289 describe the models PT(R), PT(F), and PT(C). In summary, PT(R) refers to prompt-tuning with initialization from random tokens in the vocabulary, while PT(F) pertains to initialization from a set of existing source prompts belonging to the datasets sharing the same format. On the other hand, PT(C) corresponds to the initialization involving the entire set of source prompts. We will clarify the explanations and captions further in the final camera-ready draft.

---

### Official Review · Reviewer_6o8a · 2023-08-05

**Soundness:** 5

**Excitement:**

4: Strong: This paper deepens the understanding of some phenomenon or lowers the barriers to an existing research direction.

**Paper Topic And Main Contributions:**

In this article, authors evaluate prompt tuning against full fine tuning with T5 in a transfer learning setting for Question Answering (training on multiple source dataset and applying to in-domain and out-domain QA datasets). They compare several strategies that unify the knowledge from the source datasets, and summarize the results in several key findings, comparing impact of model size and number of few shot examples.

**Questions For The Authors:**

None

**Reasons To Accept:**

This is typically the kind of paper that the NLP community needs, instead of yet another method: to a great computation cost, authors propose a comprehensive comparison of PEFT against Full Fine tuning for the unified QA problem. Even if Full Fine tuning performs best, the very few shot results of prompt tuning with correct initialization performs similarly. For QA, that requires high annotation cost, this is a key finding for real life practitioners; Additionally, the experiments on pre-training added in the appendix are also really interesting, and would be better placed in the core article (maybe the aggregation only).

The protocol is clearly presented, and the appendix seems to provide a sufficient amount of detail for reproducibility.


**Reasons To Reject:**

Only evaluated with T5.

**Reproducibility:**

4: Could mostly reproduce the results, but there may be some variation because of sample variance or minor variations in their interpretation of the protocol or method.

**Reviewer Confidence:**

3: Pretty sure, but there's a chance I missed something. Although I have a good feel for this area in general, I did not carefully check the paper's details, e.g., the math, experimental design, or novelty.

**Typos Grammar Style And Presentation Improvements:**

Including the standard deviation in the core article instead of appendix would be better, as it is a key statistic when many dataset results are aggregated. This should also be discussed in the article.

---

> ### Author Rebuttal · Authors · 2023-08-29
>
> Thank you for valuing our work.
>
> >Only evaluated with T5.
>
> While this study demonstrates outcomes using an encoder-decoder model like T5, it can easily extend to other model architectures and larger sizes, such as 6B or beyond. Due to the high computational and memory requirements for experimenting and deploying with LLMs, T5 remains a popular choice in real-world applications.  This work aims to assess the feasibility of PEFT-based techniques for unified QA. Throughout this study, open-source LLMs for prompt-tuning or fine-tuning within a reasonable computational budget were unavailable. We will leave these explorations to future work.
>
> Additionally, insights from parameter-efficient learning for medium-sized language models should guide approaches like LoRA-based fine-tuning (e.g., Llama), another widely used and parameter-efficient method. We will include the standard deviation in the core article of the camera-ready draft.

---

### Official Review · Reviewer_WoB3 · 2023-08-05

**Soundness:** 2

**Excitement:**

2: Mediocre: This paper makes marginal contributions (vs non-contemporaneous work), so I would rather not see it in the conference.

**Paper Topic And Main Contributions:**

This paper investigates unified question-answering (QA) models for low-resource scenarios and explores the potential of two tuning paradigms, model tuning, and prompt tuning, to overcome the challenges of specialized models in specific QA tasks. The study analyzes their applicability using 16 QA datasets and finds that prompt tuning can achieve comparable performance to model tuning in a few-shot setting with a good initialization.

**Questions For The Authors:**

It is not clear the definition of a task. QA itself is a task I think this paper focuses on the dataset instead of task.

**Reasons To Accept:**

1. This analysis is in-depth and comprehensive

**Reasons To Reject:**

1. Under the context of large language model, some of the motivation of this work is not valid anymore. I don't think we need to train a unified QA model by tuning the soft prompts -- the larger size of model can already answer different QA tasks well.
2. The current investigation is based on a T5-base/large model. The investigated solution is limited and there are some uncovered questions: 1) what can we expect if the LLM size goes to 13B, 65B or even larger? 2) If the model size goes larger, does the prompt tuning investigated in this work still useful?

**Reproducibility:**

3: Could reproduce the results with some difficulty. The settings of parameters are underspecified or subjectively determined; the training/evaluation data are not widely available.

**Reviewer Confidence:**

4: Quite sure. I tried to check the important points carefully. It's unlikely, though conceivable, that I missed something that should affect my ratings.

---

> ### Author Rebuttal · Authors · 2023-08-29
>
> Thank you for providing your valuable feedback.
>
> >Under the context of large language model, some of the motivation of this work is not valid anymore. I don't think we need to train a unified QA model by tuning the soft prompts -- the larger size of model can already answer different QA tasks well.
>
> Although LLMs have showcased promising adaptability across diverse tasks through in-context learning, prior research (as highlighted in https://arxiv.org/pdf/2302.06476.pdf and https://arxiv.org/pdf/2302.03494.pdf) has underscored their limitations in tackling intricate spatial, temporal, and causal common sense reasoning tasks—precisely the focus of this present study. It is apparent that when it comes to tasks involving symbolic reasoning, logical thinking, and question-answering, finetuned models outperform zero-shot LLM models by a significant margin. Notably, all of these multifaceted tasks can be framed as a question-answering task. The identified shortcomings emphasize the urgency of a model capable of effectively navigating numerous complex QA datasets, all while maintaining a competitive performance threshold within realistic computational constraints. Due to the substantial computational and memory demands associated with experimenting and deploying LLMs, T5 continues to be the favored option for real-world applications.
>
> >1) what can we expect if the LLM size goes to 13B, 65B or even larger?
>
> The findings in Table 6 indicate that prompt initialization achieves similar results to more intricate prompt-transfer methods, even when using the T5-Large model. While this research focuses on an encoder-decoder model like T5, the approach can be applied to other models like 6B or larger. The goal of this study is to assess the effectiveness of PEFT-based techniques for unified QA. During the study, no open-source LLMs were available for prompt-tuning or fine-tuning, leaving room for future investigations in these areas.
>
> >2) If the model size goes larger, does the prompt tuning investigated in this work still useful?
>
> Previous research (refer to https://arxiv.org/abs/2104.08691) indicates that fine-tuning prompts become increasingly effective as models surpass billions of parameters. Furthermore, the knowledge gained in this study from training medium-sized language models with fewer parameters can provide valuable insights for techniques such as LoRA-based finetuning, like Llama, which are well-known approaches for efficient parameter utilization.
>
> >It is not clear the definition of a task. QA itself is a task I think this paper focuses on the dataset instead of task.
>
> Although many of the existing NLP problems can be approached as question-answering tasks, we are addressing various datasets designed as question-answering tasks that emphasize different reasoning abilities, domains, and response formats. These distinct QA datasets are frequently denoted as separate tasks within the manuscript. We intend to provide further clarity on this matter in the final version of the camera-ready copy.

---

### Meta-Review · Area_Chair_4Nyg · 2023-09-23

**Recommendation:** 2

**Metareview:**

The paper investigates the question of building a unified QA model trained on a collection of QA datasets, primarily investigating if it is better to train it using full fine-tuning or prompt-tuning.

Results on T5-base and T5-large show that prompt-tuning with proper initialization for the additional param can outperform full fine-tuning.

The paper is set to explore a big question, but the execution is limited to a narrow setup that it is not clear if it generalizes. Specifically,
- Previous work showed that prompt-tuning is not flexible enough to change the model behavior significantly. The positive results with prompt-tuning might just indicate that the full-finetuning baseline wasn't trained properly
- Results in Table 5 need to be contextualized in previous results from the literature and in existing out-of-the-box methods
- Giving the known limits of prompt-tuning, other methods like lora are more promising to explore.
- Experiments are limited to T5-base and T5-large, not including auto-regressive models like opt, pythia and llama.

---

### Decision · Program_Chairs · 2023-10-07

**Decision:**

Accept-Findings

**Comment:**

The paper investigates the question of building a unified QA model trained on a collection of QA datasets, primarily investigating if it is better to train it using full fine-tuning or prompt-tuning.

Results on T5-base and T5-large show that prompt-tuning with proper initialization for the additional param can outperform full fine-tuning.

The paper is set to explore a big question, but the execution is limited to a narrow setup that it is not clear if it generalizes. Specifically,
- Previous work showed that prompt-tuning is not flexible enough to change the model behavior significantly. The positive results with prompt-tuning might just indicate that the full-finetuning baseline wasn't trained properly
- Results in Table 5 need to be contextualized in previous results from the literature and in existing out-of-the-box methods
- Giving the known limits of prompt-tuning, other methods like lora are more promising to explore.
- Experiments are limited to T5-base and T5-large, not including auto-regressive models like opt, pythia and llama.